

# RANS modelling of a single wind turbine wake in the unstable surface layer

Mads Baungaard[1], Maarten Paul van der Laan[1], and Mark Kelly[1]

[1]DTU Wind Energy, Technical University of Denmark, Risø Campus, Frederiksborgvej 399, 4000 Roskilde, Denmark

**Correspondence:** Mads Baungaard (mchba@dtu.dk)

**Abstract.** Unstable atmospheric conditions are often observed during the daytime over land and for significant periods offshore, and are hence relevant for wake studies. A simple $k$-$\varepsilon$ RANS turbulence model for simulation of wind turbine wakes in the unstable surface layer is presented, which is based on Monin-Obukhov similarity theory (MOST). The turbulence model parametrizes buoyant production of turbulent kinetic energy (TKE) without the use of an active temperature equation, and flow
balance is ensured throughout the domain by modifications of the turbulence transport equations. Large eddy simulations and experimental data from the literature are used for validation of the model.

## 1 Introduction

Wind turbine wakes have been studied for decades using many different methodologies, including wind tunnels, field experiments, analytical engineering models, and numerical simulations. A review of these methodologies is given by Porté-Agel et al.
(2020) and it is noteworthy that many of the references therein are from the past decade. The motivation for many of these new studies is the large number of new wind farms emerging each year, where wake effects significantly impact the Annual Energy Production (AEP), as well as wind farm lifetime through increased fatigue.

A sub-category of "numerical simulations" is the Reynolds-Averaged Navier Stokes (RANS) approach, which is a Computational Fluid Dynamics (CFD) method that solves for the mean fields. This means that no time history of the flow is obtained,
however the computational resources required for RANS are very small compared to higher-fidelity CFD methods, making RANS an attractive option for parametric studies or for isolating various physical effects (c.f. van der Laan et al., 2021). The wind turbine forces are commonly represented as Actuator Disks (AD) in RANS; several types of AD models are reviewed by van der Laan et al. (2015a). Compared to engineering models, an advantage of RANS is that physical features of the flow (e.g. induction zones, wake interaction, shear layers and ground effects) are solved for directly, rather than being prescribed through
empirical relations. Disadvantages are that fatigue loading can not be determined due to the steady nature of the method, and that the solution relies heavily on the turbulence model.

The part of the atmosphere closest to the ground, i.e. the atmospheric surface layer (ASL), can be parametrized with the similarity theory of Monin and Obukhov (1954) (MOST) and used as inflow for RANS simulations of wind turbine wakes. The $k$-$\varepsilon$ turbulence model is usually preferred in RANS wake studies and Crespo et al. (1985) for example utilized this (although
in a parabolized RANS setup, which requires less computational resources, but is less accurate) to simulate a single wake





in stable, neutral and unstable conditions. The wake was found to recover faster (i.e. approach the freestream velocity at a shorter downstream distance) in the unstable ASL and slower (i.e. approach the freestream velocity at a longer downstream distance) in the stable ASL compared to in a neutral ASL, and this was later confirmed in field experiments by Magnusson and Smedman (1994) and full RANS simulations, including the temperature equation, by Alinot and Masson (2002). Rados et al.

(2009) added a parametrized buoyancy term to the $k$-$\varepsilon$-equations based on the MOST expressions, eliminating the need for a temperature equation. The "indirect method" of Rados et al. was shown by El-Askary et al. (2017) to produce similar wake deficit and turbulence intensity (TI) profiles as the "direct" method that employs a temperature equation.

In all the RANS studies discussed thus far, the simulations suffer from a known imbalance in the $k$- and $\varepsilon$-equations; this means that the freestream velocity and turbulence profiles vary horizontally throughout the domain, so that different wake

results will be obtained depending on the streamwise position of the simulated turbine. van der Laan et al. (2017) solved this problem via the indirect method, by adding analytical terms to the equations, to be consistent with the ideal Tubulent Kinetic Energy (TKE) budget under MOST and thus enforce a mean balance at all points. Han et al. (2019) used this approach in the direct method, but did not show the extent to which their model is in balance.

Although there seems to be a general consensus that wakes should recover faster in unstable conditions, field measurements

by Hansen et al. (2012) and Machefaux et al. (2016) found similar wake deficits for unstable and neutral conditions. This can possibly be attributed to the large uncertainties inherent in such measurement campaigns arising from sensors, post-processing, and the unpredictable inflow provided by nature. In contrast, Large Eddy Simulation (LES) offers a controlled environment, where complete statistics of all field variables can be extracted, but at a large computational cost compared to RANS. Examples include Churchfield et al. (2012), Abkar and Porté-Agel (2015), Ghaisas et al. (2017) and Xie and Archer (2017), which

simulate wakes in both unstable and neutral conditions for a wide variety of cases. All these studies agree with the general consensus and explain it with the increased TI encountered in unstable conditions due to buoyant production of turbulence. Nevertheless, both Alinot and Masson (2002) and Keck et al. (2014) show that a faster wake recovery in unstable conditions is still observed, even when the reference TI is kept fixed (by changing the roughness length); they argue that the enhanced wake recovery must be caused by the increased turbulent length scale associated with the unstable conditions, because the turbulent

velocity scale is approximately constant for fixed reference TI and wind speed.

The balanced $k$-$\varepsilon$ MOST model by van der Laan et al. (2017) can be combined with the $f_P$-correction, which was originally formulated by Apsley and Castro (1997) and later used by van der Laan (2014) to circumvent the over-diffusiveness of the $k$-$\varepsilon$ model in the wake region. However, the $f_P$-limiter was derived and calibrated for a neutral ASL, where it has been applied in many cases with success, but for non-neutral conditions it has yielded unphysical behaviour, especially in unstable cases

(van der Laan et al., 2021). Modifications to the MOST $k$-$\varepsilon$-$f_P$ equations in the unstable regime are therefore suggested in this paper and validated against various field experiments and LES's.





## 2 Simulation setup

The wakes are simulated with the incompressible, finite-volume flow solver EllipSys3D (Michelsen, 1992; Sørensen, 1995). After continuous development since then, it is now a highly scalable code, which can be run in parallel on large high perfor-
mance clusters (HPC) via Message Passing Interface (MPI). Thus a typical RANS simulation of a single wake only takes a few minutes to simulate on a contemporary HPC, while a similar case with LES would take several hours to run, even with an order of magnitude more computer resources available. In terms of CPU-hours, van der Laan et al. (2015b) estimated LES to be approximately $10^3$ times more expensive than RANS and this estimate may even be considered conservative, because the LES inflow was created using a Mann-model turbulence box, and not with the more expensive precursor method. Furthermore,
several LES runs are in principle necessary in order to create an ensemble average, which multiplies the cost of LES. This clearly motivates the development of the RANS model as a fast, albeit less accurate, alternative to LES.

The different components of the RANS simulations will be discussed in the following sections.

### 2.1 Inflow profile for unstable ASL

Numerous articles have been written about MOST and a historical review is given by Foken (2006). The theory is expressed
and applied via the dimensionless stability parameter

$$\zeta \equiv \frac{z}{L}, \tag{1}$$

where $z$ is the height above ground and $L$ is the Obukhov length. Negative $\zeta$ corresponds to unstable conditions, while $\zeta = 0$ corresponds to the neutral limit where there is no effect of buoyancy. Neutral conditions are typically defined as $|L|^{-1} \lesssim 0.002 \text{ m}^{-1}$ (e.g. Gryning et al., 2007) and tend to occur most often, with observed distributions of the stability ($1/L$ or $\zeta$)
having a peak around zero (Kelly and Gryning, 2010). The most common unstable Obukhov lengths occur at $-0.02 \text{ m}^{-1} \lesssim L^{-1} \lesssim -0.002 \text{ m}^{-1}$ (Kelly and Gryning, 2010); but offshore, there tends to be a bias towards more unstable conditions, i.e. more negative $L^{-1}$ compared to onshore (Sathe et al., 2013). Various parametrizations have been suggested for wind speed, $k$, and $\varepsilon$ in terms of $\zeta$; in this paper we use the widely accepted forms of Dyer (1974) for $U$ (namely the $\Psi_m$ and $\Phi_m$ functions), and those found in Kaimal and Finnigan (1994) for $\varepsilon$ and $k$ (see van der Laan et al., 2017, for more details). Under unstable
conditions these are:

$$U = \frac{u_*}{\kappa}\left[\ln\left(\frac{z}{z_0}\right) - \Psi_m\right] \quad , \quad V = W = 0, \tag{2}$$

$$k = \frac{u_*^2}{\sqrt{C_\mu}}\left(\frac{\Phi_\varepsilon}{\Phi_m}\right)^{1/2}, \tag{3}$$

$$\varepsilon = \frac{u_*^3}{\kappa z}\Phi_\varepsilon, \tag{4}$$





where

$$\Psi_m = \ln\left[\frac{1}{8}\left(1 + \Phi_m^{-2}\right)\left(1 + \Phi_m^{-1}\right)^2\right] - 2\arctan\left(\Phi_m^{-1}\right) + \frac{\pi}{2},\tag{5}$$

$$\Phi_m = (1 - 16\zeta)^{-1/4},\tag{6}$$

and $\hspace{12cm}$ (7)

$$\Phi_\varepsilon = 1 - \zeta.\tag{8}$$

The above relations are valid for $-2 \lesssim \zeta < 0$, so for a fixed $L < 0$, it means that the equations are in principle only valid

up to $z \approx -2L$, where the free convection regime starts (i.e. buoyant production dominates over shear production of TKE). Although the blade tip of a modern turbine can reach beyond $-2L$ in unstable conditions (e.g. for $z_{\text{tip}} = 200\,\text{m}$ this happens when $L^{-1} \lesssim -0.01\,\text{m}^{-1}$, which is not rare), we nevertheless still choose to apply the profiles—and in fact use them all the way up to the top boundary. More realistic inflow profiles for RANS covering the whole Atmospheric Boundary Layer (ABL) are indeed a current research topic, but will not be discussed further in this paper. Maronga and Reuder (2017) reason that MOST

is a "pragmatic solution," because the parameters needed for more realistic inflow profiles are often not available.

In this paper $\kappa = 0.4$ and $C_\mu = 0.03$ are used, while $z_0$ and $u_*$ in Eqs. (2) to (8) can be set using reference values (i.e. defined at $z = z_{\text{ref}}$) of wind speed ($U_{\text{ref}}$) and total TI ($I_{\text{ref}}$) along with the stability parameter ($\zeta_{\text{ref}}$):

$$z_0 = z_{\text{ref}}\exp\left[-I_{\text{ref}}^{-1}C_\mu^{-1/4}\kappa\sqrt{\frac{2}{3}}\left(\frac{\Phi_\varepsilon(\zeta_{\text{ref}})}{\Phi_m(\zeta_{\text{ref}})}\right)^{1/4} - \Psi_m(\zeta_{\text{ref}})\right]\tag{9}$$

$$u_* = U_{\text{ref}}I_{\text{ref}}C_\mu^{1/4}\sqrt{\frac{3}{2}}\left(\frac{\Phi_m(\zeta_{\text{ref}})}{\Phi_\varepsilon(\zeta_{\text{ref}})}\right)^{1/4}.\tag{10}$$

Note that TI ($I$) here is not the same as streamwise turbulence intensity ($\sigma_u/U$); in this paper "TI" will refer to total TI (i.e., $I \equiv \sqrt{\frac{2}{3}k}/U$), unless stated otherwise. A typographical (sign) error has been corrected in Eq. (9), compared to the similar expression found in van der Laan et al. (2017).

Examples of four inflow profiles with identical $U_{\text{ref}}$ are shown in Fig 1. The stability and TI differ among the cases, but they

still have approximately the same averaged power density; i.e., $\frac{1}{A}\iint\frac{1}{2}\rho U(z)^3 dA \approx 308.9\,\text{W m}^{-2}$ ($\pm 0.6\,\%$). One peculiarity of the unstable MOST profiles is that the TI does not go to zero for $z \to \infty$; this is connected to the ASL assumptions used by MOST (i.e., the mixed and upper layers above the ASL are lacking, where $I$ becomes constant and then vanishes). Another peculiarity (for both neutral and unstable conditions) is that higher TI leads to larger shear ($dU/dz$), because the velocity gradient scales with $u_*$, which scales with $I_{\text{ref}}$ (Eq. 10); this is a consequence of specifying both TI and hub height velocity.

The eddy viscosity profile, $\nu_t(z) = C_\mu\frac{k^2}{\varepsilon}$, is especially interesting to compare between the cases, because $\nu_t$ features as a diffusion coefficient in the Reynolds-Averaged momentum equation and therefore is connected with the entrainment of ambient air into the wake. A faster wake recovery is therefore expected for larger $\nu_t$, which as seen in Fig. 1 can be obtained by increasing either the turbulence strength ($I_{\text{ref}}$), or how unstable the atmosphere is ($-\zeta_{\text{ref}}$), or both.



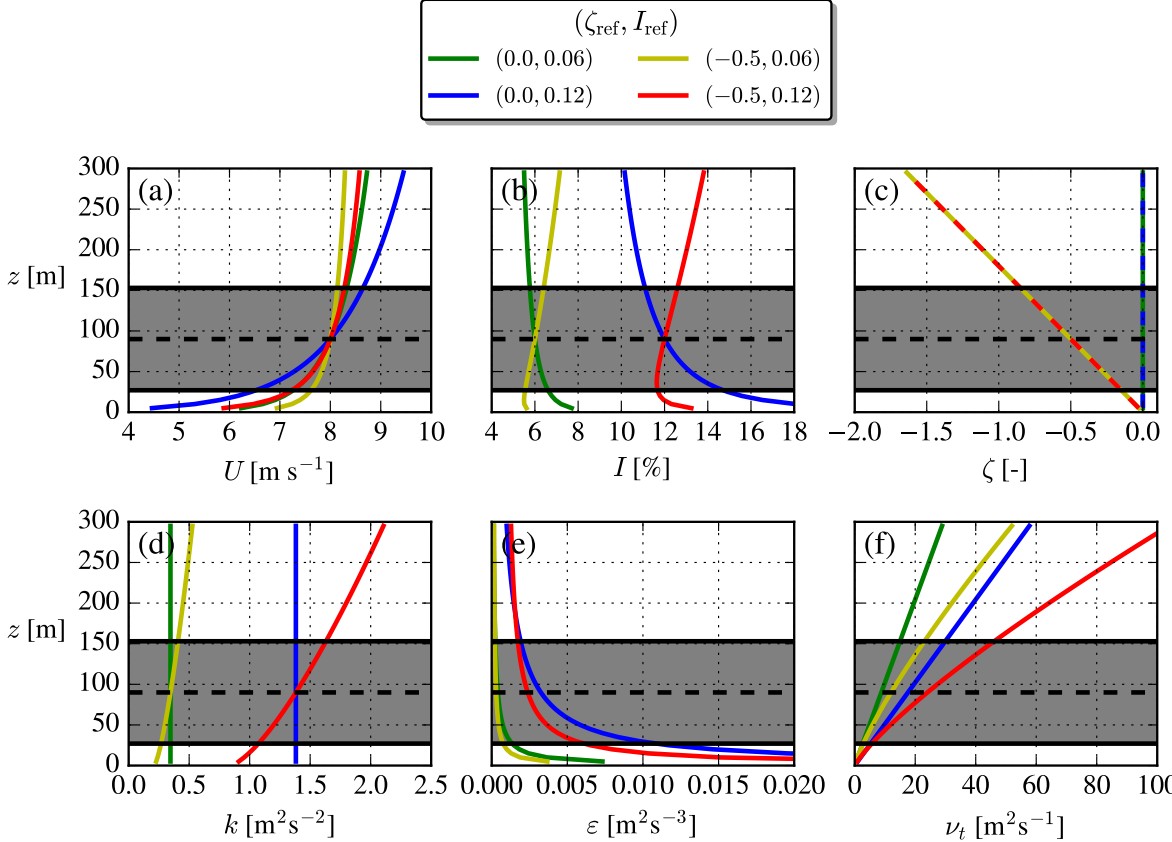

**Figure 1.** Analytical MOST profiles. Combinations of low/high TI and neutral/unstable stability. The rotor area of a NREL5MW turbine is shown. Dashed lines are used for $\zeta$ to make all profiles visible.

The eddy viscosity is sometimes expressed as a product of turbulent velocity and length scales, c.f. Pope (2000): $\nu_t = u_t \ell_t$,

where $u_t = C_\mu^{1/4} k^{1/2}$ and $\ell_t = C_\mu^{3/4} k^{3/2} \varepsilon^{-1}$. These are plotted in Fig. 2, from which it is clear that TI only affects $u_t$ ($l_t$ is unaffected due to $u_*$ cancelling when dividing $k^{3/2}$ from Eq. (3) by $\varepsilon$ from Eq. (4)), while stability mainly alters $\ell_t$.



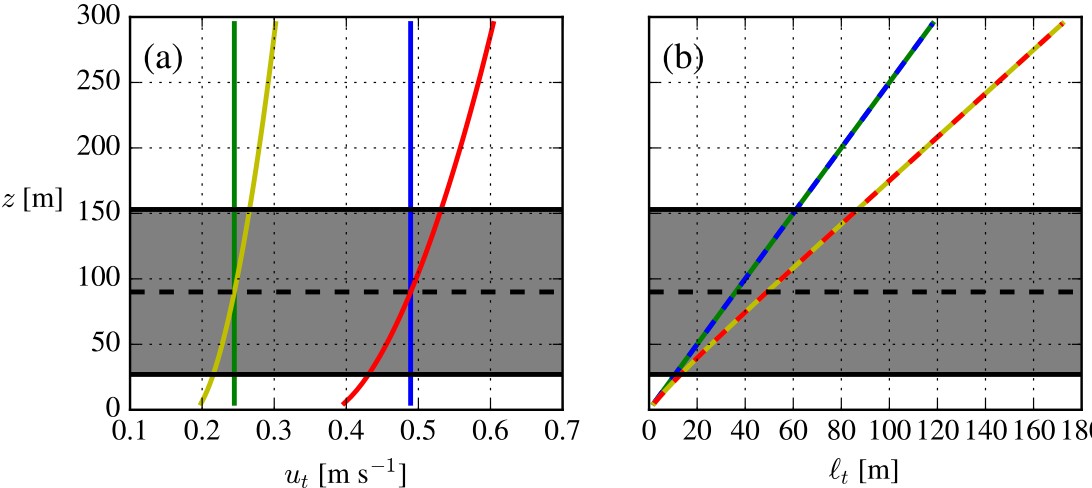

**Figure 2.** Turbulence scales in the freestream. Same labels as in Fig. 1.

## 2.2 Wind Turbine representation

A recently developed Actuator Disk (AD) model by Sørensen et al. (2020) is utilized in this paper. The model can be derived from conservation of energy (Bernoulli's equation), conservation of angular momentum (Euler's turbine equation) and an analytical expression for the near-wake azimuthal velocity distribution. The latter is modelled by a vortex extending from the center of the AD to infinity with constant circulation, hence it resembles the classical Joukowsky optimum rotor, c.f. Okulov and Sørensen (2010), and the AD model is therefore referred to as the "Joukowsky-AD". A summary of the model formulation is given in Appendix A2.

The main advantage of the Joukowsky-AD over the widely used "airfoil-AD" (e.g. Sørensen and Kock (1995), Porté-Agel et al. (2011) and van der Laan et al. (2015a)) is that only a few parameters are necessary: The thrust coefficient $C_T$, tip-speed ratio $\lambda$, rotor radius $R$ and freestream reference wind speed $U_{\text{ref}}$ (in addition to these, the power coefficient $C_P$ is also made an input parameter in our implementation as described by van der Laan et al. (2020)). Nevertheless, it is still able to model non-axissymmetric force distributions and wake rotation, similar to the disk loading of an airfoil-AD. Porté-Agel et al. (2011) argued that these are important features to capture the correct wake behaviour in the near-wake, while van der Laan et al. (2015c) showed that they are only of minor importance for the far-wake. Wake deficit and rotor loading of the Joukowsky-AD have been found to compare well with several validation cases conducted by Sørensen et al. (2020) and Sørensen and Andersen (2020). This is verified to also be the case for our RANS simulations in Appendix A2.

No nacelle nor tower are included in our simulations, which have been shown to be a good approximation for $> 3D$ downstream of the turbine according to Kasmi and Masson (2008) and Li et al. (2020).



## 2.3 RANS

A homogeneous, flat lower surface is assumed for all cases in this paper. The inner part of the mesh surrounding the AD is called the "wake domain" and is shown for a typical case in Fig. 3. In this area, a horizontal resolution of $\Delta x = \Delta y = D/10$ is used (based on the grid study in Appendix A1), while grid stretching is used in the vertical direction with $\Delta z = z_0$ at the first cell and $\Delta z = D/10$ at the cell at $z/D = 3$. The wake domain is however only a small part of the full domain: The full domain extends an additional $x_{in} = 5$ km to the west, north, east and south, respectively, while the top of the grid is at $z/D = 25$. Grid stretching is used in all directions outside of the wake domain to circumvent an excessively large number of cells (the case shown in Fig. 3 has $\approx 2.1 \cdot 10^6$ cells in total). The choice of having grids with size on the order of $\sim 10$ km is made to avoid tunnel-like blockage effects, and to have fully developed inflow profiles at the turbine position.

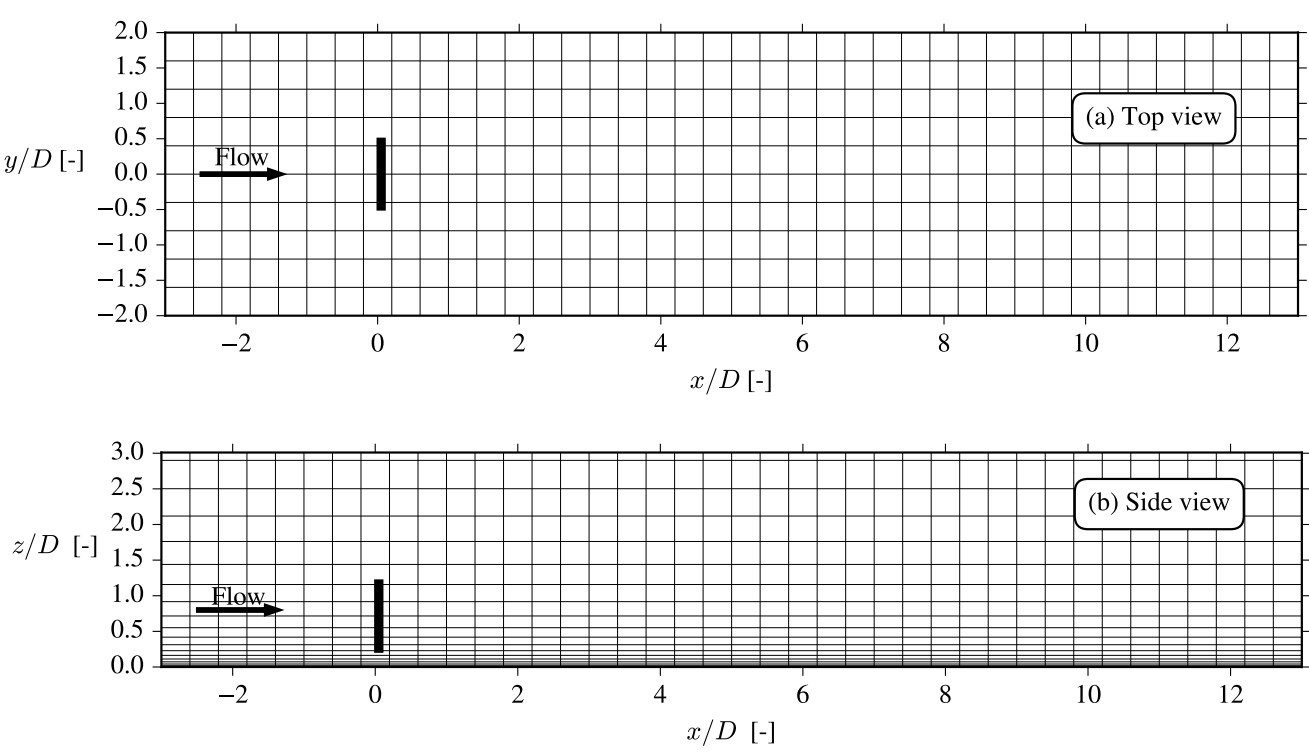

**Figure 3.** Top and side views of the wake domain, which size is $\{l_x, l_y, l_z\}/D = \{16, 4, 3\}$. The total grid size is $\{L_x, L_y, L_z\} = \{l_x + 2x_{in}, l_y + 2x_{in}, l_z + 22D\}$ and is too large to be shown here, because $x_{in} = 5$ km. Only every 4'th cell is plotted.

The numerical solution strategy of the incompressible RANS equations in EllipSys3D is thoroughly discussed in other publications (Michelsen, 1992; Sørensen, 1995; Sørensen et al., 2007), so only the main features are discussed here. The SIMPLE method is used with a modified Rhie-Chow algorithm, following Réthoré (2009) and Troldborg et al. (2015), to avoid the numerical wiggles induced by the discrete actuator disk body forces.





As mentioned in the introduction, the flow variables in an empty domain with MOST inflow can be kept in balance by modifying the $k$- and $\varepsilon$-equations, as suggested by van der Laan et al. (2017):

$$U_j \frac{\partial k}{\partial x_j} = \mathcal{D}_k + \mathcal{P} - \varepsilon + \mathcal{B} - S_k, \tag{11}$$

$$U_j \frac{\partial \varepsilon}{\partial x_j} = \mathcal{D}_\varepsilon + (C_{\varepsilon 1}\mathcal{P} - C_{\varepsilon 2}\varepsilon + C_{\varepsilon 3}\mathcal{B}) \frac{\varepsilon}{k}, \tag{12}$$

where

$$\mathcal{D}_{\{k,\varepsilon\}} = \frac{\partial}{\partial x_j} \left( \frac{\nu_t}{\sigma_{\{k,\varepsilon\}}} \frac{\partial \{k,\varepsilon\}}{\partial x_j} \right) \tag{13}$$

$$\mathcal{P} = -\overline{u_i' u_j'} \frac{\partial U_i}{\partial x_j} \tag{14}$$

$$\overline{u_i' u_j'} = \frac{2}{3} k \delta_{ij} - \nu_t \left( \frac{\partial U_i}{\partial x_j} + \frac{\partial U_j}{\partial x_i} \right) \tag{15}$$

$$\nu_t = C_\mu f_P \frac{k^2}{\varepsilon} \tag{16}$$

$$S_k = \frac{u_*^3}{\kappa L} \left[ \zeta^{-1}(\Phi_m - \Phi_\varepsilon) - 1 - \frac{\kappa^2}{4\sigma_k \sqrt{C_\mu}} \Phi_m^{13/2} \Phi_\varepsilon^{-3/2} f_{un} \right] \tag{17}$$

$$f_{un} = (2 - \zeta) + 8(1 - 12\zeta + 7\zeta^2) - \frac{1}{16}(3 - 54\zeta + 35\zeta^2) \tag{18}$$

$$C_{\varepsilon 3} = \frac{1}{\zeta} \left( C_{\varepsilon 1}\Phi_m - C_{\varepsilon 2}\Phi_\varepsilon + [C_{\varepsilon 2} - C_{\varepsilon 1}]\Phi_\varepsilon^{-1/2}\Phi_m^{5/2}(1 - 12\zeta) \right). \tag{19}$$

The source term, $S_k$, and the $C_{\varepsilon,3}$ parameter constitute the two modifications compared to the usual $k$-$\varepsilon$ equations (similar corrections exist for the stable ASL, but are not discussed in this paper). Viscous terms have been neglected in the above equations, which is a good approximation in atmospheric flow applications due to the Reynolds number being very large (Wyngaard, 2010). The Coriolis force is also neglected, hence no veer is present in the simulations. Definitions of the $f_P$-correction (which was in fact set to $f_P = 1$ in the work of van der Laan et al. (2017)) and buoyant production, $\mathcal{B}$, are deferred to the Section 3. The parameters used in the above equations are summarized in Table 1. Finally, $S_k$ and $C_{\varepsilon,3}$ differ slightly from those printed in van der Laan et al. (2017), with the only difference being that here we choose $\frac{\Phi_h}{\Phi_m \sigma_\theta} \to 1$; this "modeller's choice" for turbulent Prandtl number ($\sigma_\theta$) avoids the inconsistency mentioned in that paper, and makes the model independent of the temperature similarity function $\Phi_h$.

| $C_{\varepsilon,1}$ | $C_{\varepsilon,2}$ | $\sigma_k$ | $\sigma_\varepsilon$ | $C_\mu$ | $\kappa$ |
|---|---|---|---|---|---|
| 1.21 | 1.92 | 1.00 | 1.30 | 0.03 | 0.4 |

**Table 1.** Parameters of the $k$-$\varepsilon$ MOST turbulence model.





## 3 Modification of the $k$-$\varepsilon$-$f_P$ model in the unstable ASL

The background eddy viscosity shown in Fig. 1 is perturbed in the turbine presence and especially so in the wake region, see Fig. 4 for an example with neutral inflow. As explained in Section 2.1, $\nu_t$ is very important for the wake development and the $f_P$-correction effectively attenuates the $\nu_t$ perturbation in the interface between the wake and freestream, known as the shear layer, and in the region around the AD to improve wake predictions. This attenuation can also be viewed as a modification of the turbulence scales, $u_t$ and $\ell_t$.

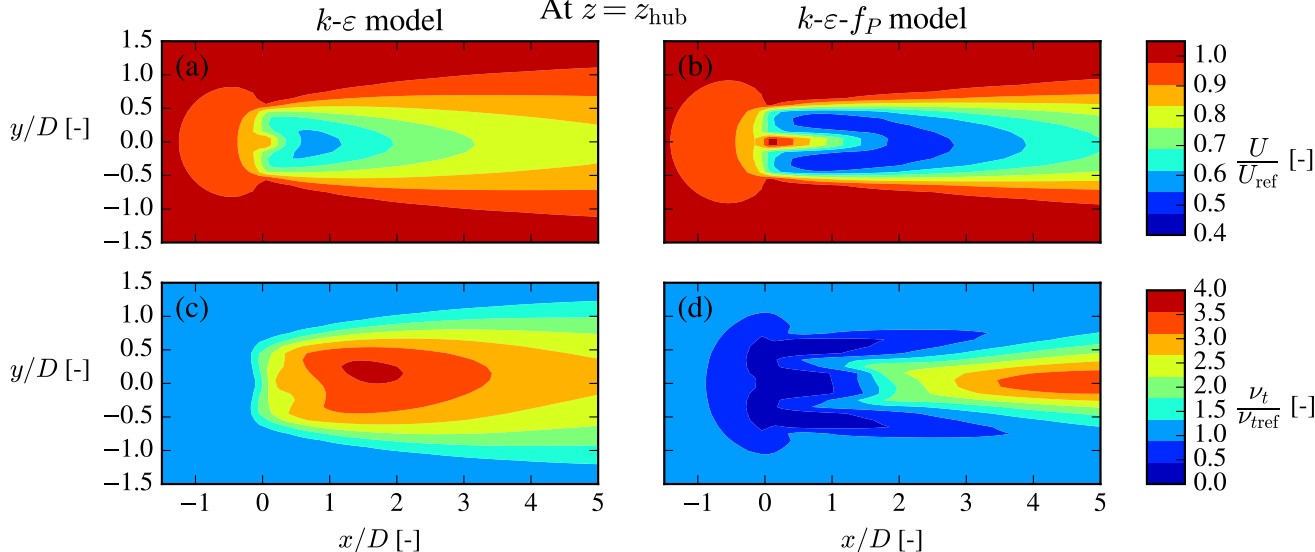

**Figure 4.** Streamwise velocity (upper row) and kinematic eddy viscosity (lower row) are normalized by their freestream values. The neutral $I_{\mathrm{ref}} = 6\,\%$ case from Fig. 1 with a single NREL5MW turbine is shown.

The cause of the $\nu_t$ perturbation in the first place is the large velocity gradients across the AD and the shear layer, which enhances TKE shear production, but other terms of Eq. (11) are also highly active in these regions, and it is this complex interplay together with the $f_P$-formulation that in the end determine the wake recovery. The effect of the buoyancy term in this interplay is discussed first, then afterwards the role of $f_P$ in the unstable ASL.

### 3.1 Buoyant production term

The buoyant production of TKE is $\mathcal{B} \equiv \frac{g}{\theta_0} \overline{w'\theta'}$ and the heat flux is typically obtained using a temperature equation and a flux-gradient relationship. In this work, we pursue an alternative way and investigate two simple parametrizations:





$$\mathcal{B} = -\nu_t \left[ \left( \frac{\partial U}{\partial z} \right)^2 + \left( \frac{\partial V}{\partial z} \right)^2 \right] \frac{\zeta \Phi_h}{\sigma_\theta \Phi_m^2} \quad \text{(2017 model)} \tag{20}$$

$$\mathcal{B} = -\frac{u_*^3}{\kappa L} \quad \text{(cstB model)} \tag{21}$$

The "2017 model" is the one utilized by van der Laan et al. (2017), van der Laan et al. (2020), Doubrawa et al. (2020) and

van der Laan et al. (2021). This model does not require a temperature equation for closure, but instead utilizes the temperature similarity function, $\Phi_h$ and Prandtl number, $\sigma_\theta$. The "cstB" model is as the name suggests simply a constant source term throughout the domain and again no temperature equation is necessary. The $k$- and $\varepsilon$-equations are the same for the two models, except some minor changes to $S_k$, Eq. (17), and $C_{\varepsilon 3}$, Eq. (19), are needed for the 2017 model, see Section 2.3. To isolate the effect of the $\mathcal{B}$ parametrizations, they are tested here with $f_P = 1$. A NREL5MW turbine (Jonkman et al., 2009)

with $U_{\text{ref}} = 8 \text{ m/s}$ is used for all plots in this section.

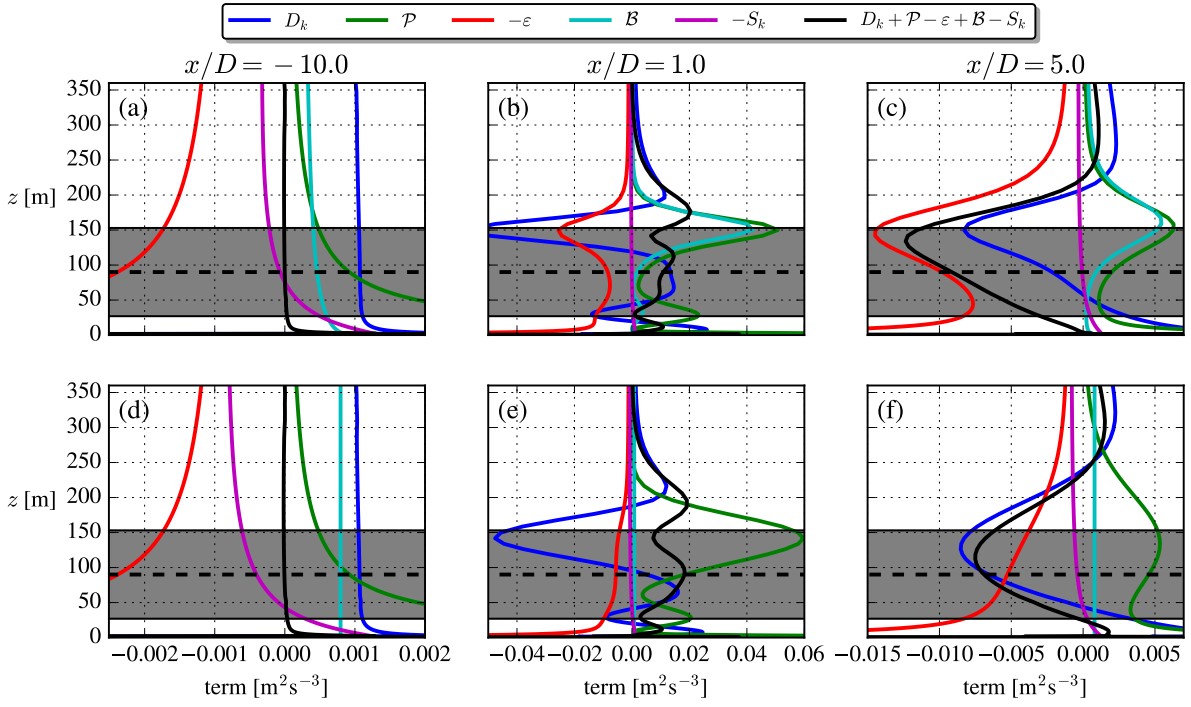

**Figure 5.** TKE budgets of the 2017 model (upper row) and cstB model (lower row). The profiles are extracted at $y/D = 0$, i.e. in the center of the wake. $\zeta_{\text{ref}} = -0.5$ and $I_{\text{ref}} = 12 \%$ for both rows.

The upstream ($x/D = -10$) budget in Fig. 5 shows the "inconsistency" of the 2017 model mentioned by van der Laan et al. (2017): The buoyant production goes to $\mathcal{B} = \frac{-u_*^3}{\kappa L} \Phi_m$, although $\mathcal{B} \rightarrow \frac{-u_*^3}{\kappa L}$ is expected in the freestream, which can be derived from the ASL definition $\mathcal{B} \equiv \frac{g}{\theta_0} \overline{\theta' w'}_s$ and the Obukhov length definition. The cstB model on the other hand by definition





complies with the freestream ASL limit of $\mathcal{B}$. Additionally, it can be noted that the cstB model has $\mathcal{B}/\mathcal{P} \approx 1$ at $z_{\mathrm{ref}}$, because

$\zeta_{\mathrm{ref}} = -0.5$ was used (c.f. Fig. 5.23 of Stull, 1988).

A clear distinction between the two parametrizations are seen both in the near-wake ($x/D = 1$) and far-wake ($x/D = 5$) TKE budgets: In the top shear layer of the 2017 model simulation, large buoyant production is produced by the large velocity gradients in this region. This is neither observed in direct RANS simulations by El-Askary et al. (2017) nor in wind tunnel experiments by Zhang et al. (2013) or Hancock and Zhang (2015), so this may be deemed as an unphysical artifact. Indeed the

2017 model is derived for a homogeneous ASL and applying it to a wind turbine wake violates this assumption. The cstB model on the other hand effectively uncouples the buoyant production and wake dynamics. This assumption can partly be justified by the aforementioned studies, which show that temperature changes very little in the wake from the ambient conditions and that the heat flux actually *decreases* in the wake.

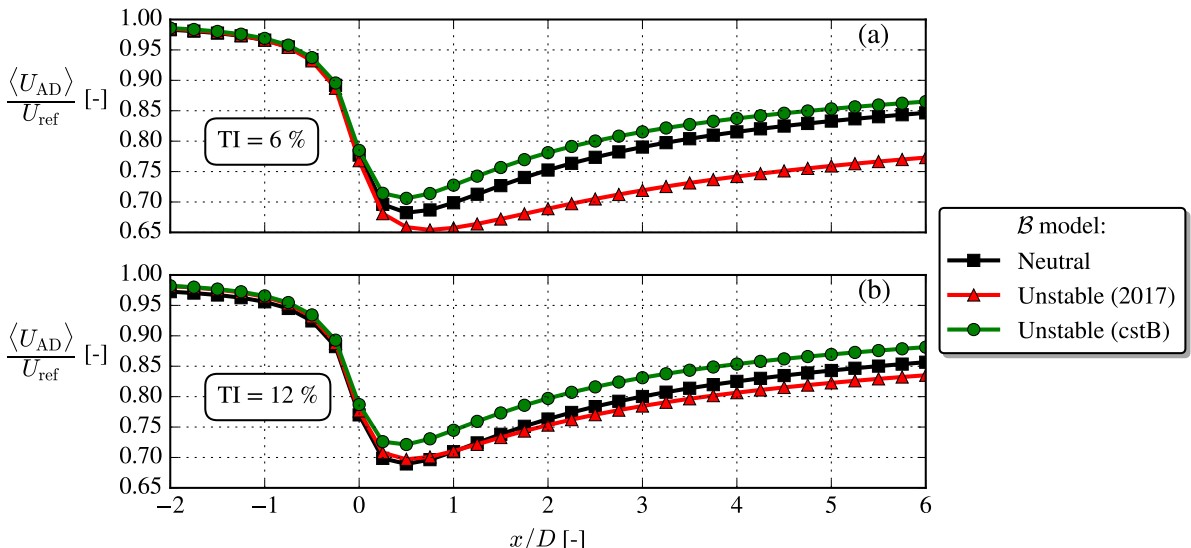

**Figure 6.** Disk averaged streamwise velocity, $\langle U_{\mathrm{AD}} \rangle$, for low/high TI and neutral/unstable. $\zeta_{\mathrm{ref}} = -0.5$ for unstable.

Another deficiency of the unstable 2017 model is illustrated in Fig. 6: For a given $I_{\mathrm{ref}}$, it unphysically predicts slower wake

recovery than in neutral as also noted by van der Laan et al. (2021). This is remedied in the cstB model, where a slightly faster wake recovery is seen. It can be noted that in the near- and far-wake of the cstB model both $\mathcal{B}$ and $S_k$ terms are small compared to the other TKE terms, c.f. Fig 5, and as such it effectively resembles the neutral model, but with the one difference that it has a larger turbulent length scale, c.f. Fig. 2, which explains the faster wake recovery seen in Fig. 6. The $\mathcal{B}$ and $S_k$ terms must nevertheless still be retained to enforce the freestream balance of the $k$- and $\varepsilon$-equations throughout the domain.

The faster wake recovery of the unstable cstB model compared to the similar neutral case is also seen in Fig. 7 along with the shear parameter and turbulence time scale, which are both used in the $f_P$-correction to be discussed in the next section.



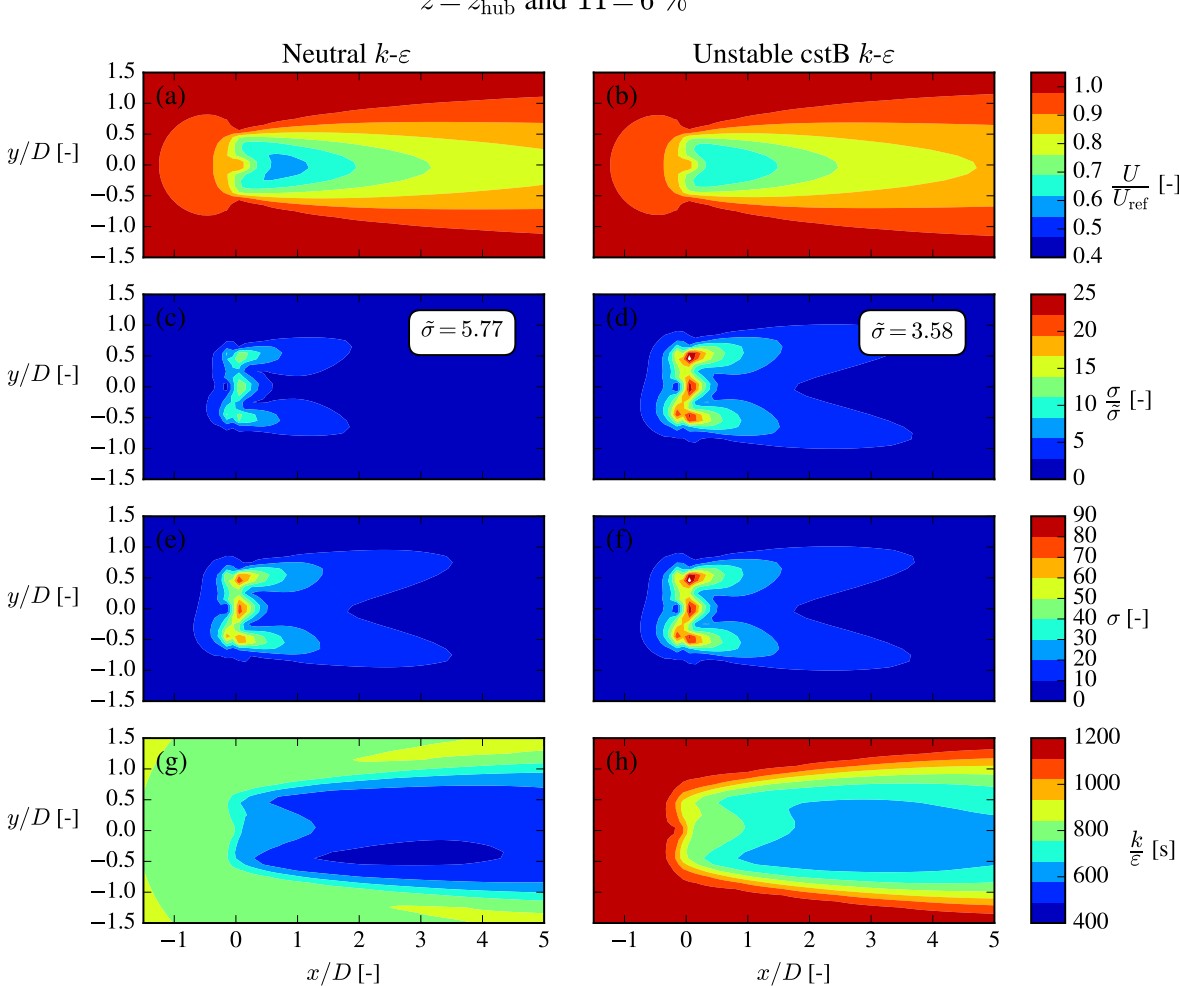

**Figure 7.** Normalized streamwise velocty (1st row) , normalized shear parameter (2nd row), shear parameter (3rd row) and turbulence time scale (4th row). The cstB model is used for the unstable case (right column), which has $\zeta_{\mathrm{ref}} = -0.5$.

## 3.2 Turbulence closure with $f_P$ in non-neutral conditions

As stated in the introduction, $k$-$\varepsilon$ models tend to predict faster wake recovery compared to experiments and LES. This can be corrected by using $f_P \neq 1$ in the $\nu_t$ definition, Eq. (16), which clearly affects the velocity deficit as shown in Fig. 4. The form





of $f_P$ used for wakes in the neutral ASL by van der Laan (2014), can be summarized as:

$$f_P = \frac{2f_0}{1 + \sqrt{1 + 4f_0(f_0 - 1)(\sigma/\tilde{\sigma})^2}} \tag{22}$$

$$f_0 = 1 + \frac{1}{(C_R - 1)} \quad \text{(neutral)} \tag{23}$$

$$\sigma = \frac{k}{\varepsilon}\sqrt{\left(\frac{\partial U_i}{\partial x_j}\right)^2} \tag{24}$$

$$\tilde{\sigma} = \frac{1}{\sqrt{C_\mu}} \quad \text{(neutral)}. \tag{25}$$

The shear parameter, $\sigma$, is large in the region surrounding the rotor and in the shear layer, see Fig. 7, which explains the drop
of $\nu_t$ in these regions.

    It was recognized by van der Laan et al. (2020), that the freestream shear parameter, $\tilde{\sigma}$, has to be adjusted for MOST inflow
in order to have $f_P = 1$ in the freestream:

$$\tilde{\sigma} = \frac{1}{\sqrt{C_\mu}}\sqrt{\frac{\Phi_m}{\Phi_\varepsilon}}. \quad \text{(general)} \tag{26}$$

This can simply be derived by inserting the freestream profiles of $U$, $k$ and $\varepsilon$ (see Sect. 2.1) into the shear parameter definition,
Eq. (24); the form of Eq. (26) has been used in all previous papers utilizing the 2017 model for wake modelling.

    A more subtle modification arises recognizing that the $f_0$ parameter is also stability-dependent, i.e.,

$$f_0 = 1 + \frac{C_\mu \tilde{\sigma}^2}{C_R - 1} \quad \text{(modification 1)}. \tag{27}$$

This is actually the form suggested by Apsley and Leschziner (1998), but they were not considering stability effects, i.e. no
variation of $\tilde{\sigma}$ nor $C_R$ with stability; it has not been used in previous applications of the 2017 model. We note Eq. (27) is
consistent with the neutral limit, since $\tilde{\sigma}^2 \to C_\mu^{-1}$ for $\zeta \to 0$. The Rotta constant was calibrated to $C_R = 4.5$ for wind turbine
wakes in the neutral ASL in the work of van der Laan (2014) and we therefore require $C_R \to 4.5$ in the neutral limit ($\zeta \to 0$).
One form that satisfies this is

$$C_R = 4.5 + C_B \frac{\tilde{B}}{\tilde{\varepsilon}} \quad \text{(modification 2)}, \tag{28}$$

where $\tilde{B}/\tilde{\varepsilon}$ is the freestream buoyant production to dissipation ratio and $C_B$ is a new parameter to be calibrated. The effect of
modifications 1 and 2 is shown in Fig. 8. It is clear that the two modifications increase $f_P$ in regions where $\sigma/\tilde{\sigma} > 1$, i.e. in the
wake. This is necessary because larger $\sigma/\tilde{\sigma}$ are encountered with non-neutral inflow, since $\tilde{\sigma}$ decreases and $k/\varepsilon$ increases in
unstable conditions (note from Eq. (24) that $\sigma \sim k/\varepsilon$); see Fig. 7.

    When both modifications are used, faster wake recovery for a given $I_{\text{ref}}$ occurs in unstable conditions, as shown in Fig. 9;
this was also the case when $f_P$ was 'turned off' (set to 1), c.f. Fig. 6.



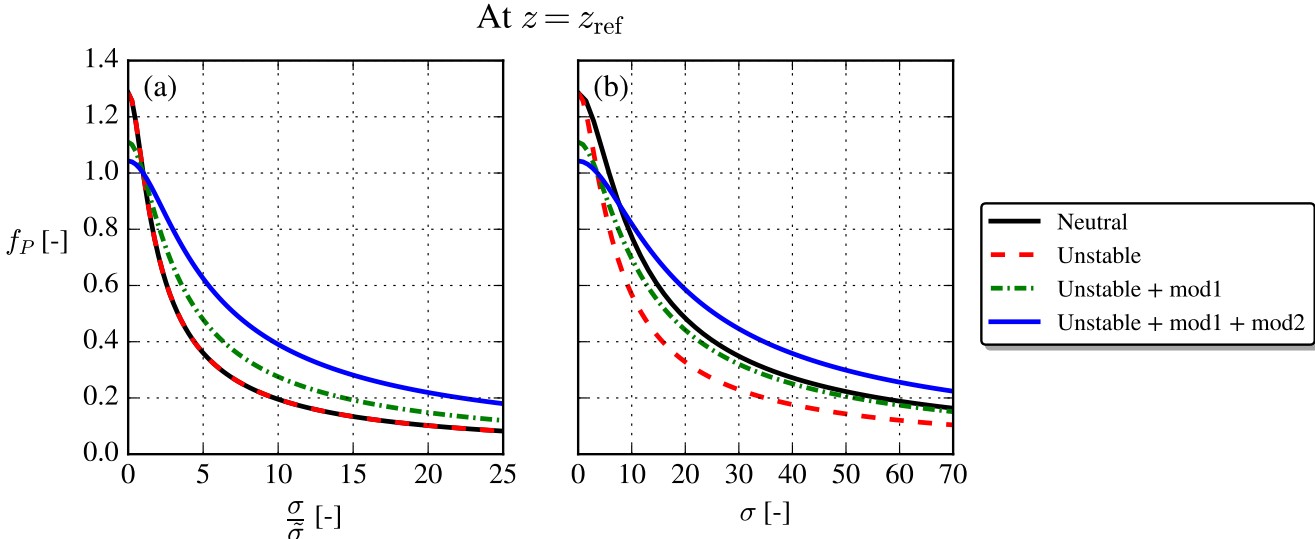

**Figure 8.** The $f_P$-correction as function of normalized shear parameter and shear parameter, respectively, for $\zeta_{\mathrm{ref}} = -0.5$ (it is independent of TI). The effect of modification 1 (Eq. 27) and modification 2 (Eq. 28 with $C_B = 10.0$) are shown.

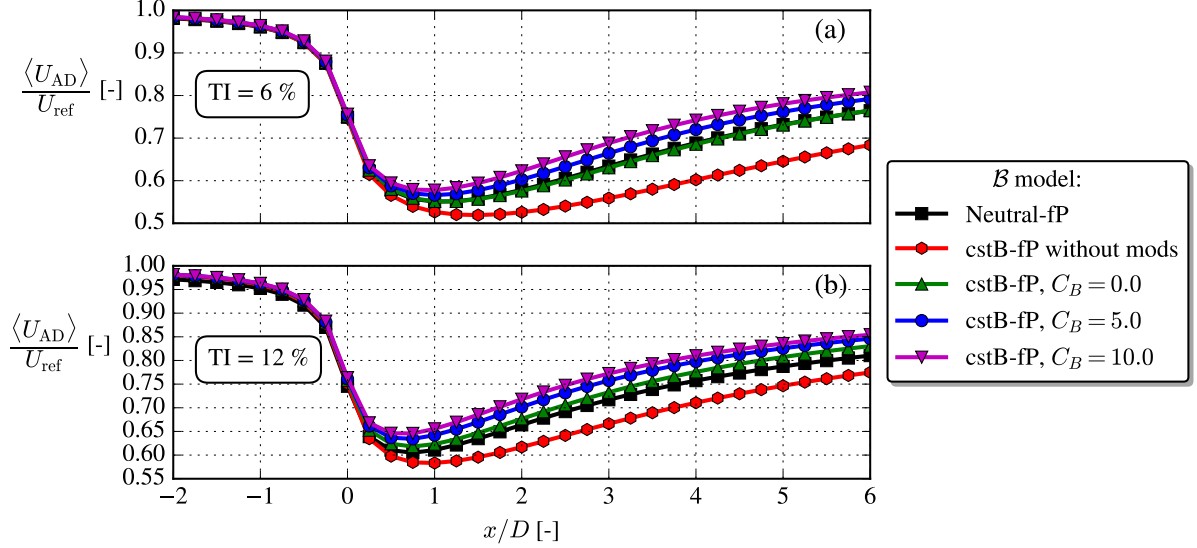

**Figure 9.** Same as Fig. 6, but with $f_P$ included. $C_B = \{0.0, 5.0, 10.0\}$ are tested for the unstable cstB model.





# 4 Validation with experiments and LES

The cstB model with the $f_P$-modifications described in the previous section is tested for the cases summarized in Table 2. Each case is simulated with a range of $C_B$ parameters, $C_B = \{0.0, 5.0, 10.0\}$, while keeping $C_R = 4.5$ fixed. The latter has been calibrated for a suite of neutral EllipSys3D LES's by van der Laan (2014), but in practice if it was calibrated with another LES code, a different, optimal $C_R$ might have been obtained. In the same way, it cannot be expected that a universally valid $C_B$ exists, when we compare with results from a range of different codes and experiments. Hence, no optimal $C_B$ will be obtained in this section, but rather the qualitative effect of $C_B$ is shown.

The numerical setup for each case follows that described in Section 2.3, e.g. the cell size and extent of the wake region are scaled with the rotor diameter.

| Case | Type | $D$ [m] | $z_{\mathrm{ref}}$ [m] | $C_T$ [-] | $P$ [kW] | $\Omega$ [rpm] | $U_{\mathrm{ref}}$ [m/s] | $I_{\mathrm{ref}}$ [%] | $\zeta_{\mathrm{ref}}$ [-] |
|---|---|---|---|---|---|---|---|---|---|
| SWiFT | LES, Exp. | 27 | 32.1 | 0.81 | 52 | 37.0 | 6.7 | 10.0 | -0.29 |
| NTK41 | LES, Exp. | 41 | 36 | 0.83 | 125 | 27.1 | 6.8 | 15.0 | -0.42 |
| V80-Abkar | LES | 80 | 70 | 0.81 | 696 | 16.1 | 8.0 | 8.1 | -0.47 |
| V80-Keck | LES | 80 | 70 | 0.81 | 696 | 16.1 | 8.0 | 6.1 | -0.84 |
| NREL5MW | LES | 126 | 90 | 0.77 | 1808 | 9.1 | 8.0 | 7.0 | $-1.32$ |

**Table 2.** Overview of testcases. SWiFT: Doubrawa et al. (2020). NTK41: Machefaux et al. (2016). V80-Abkar: Abkar and Porté-Agel (2015). V80-Keck: Keck et al. (2014). NREL5MW: Churchfield et al. (2012). The air density used for all cases is $\rho = 1.225$ kg m$^{-3}$.

## 4.1 SWiFT case

A large wake benchmark study was conducted by Doubrawa et al. (2020) to compare various simulation methodologies and codes against LIDAR measurements in different atmospheric conditions. The measurements were carried out for a Vestas V27 turbine at the Scaled Wind Technology Facility (SWiFT) in Lubbock, Texas, USA, which is an area of flat terrain.

The inflow parameters of the SWiFT row in Table 2 were obtained from the ensemble average of five 10 min-averages from a met. mast located $2.5D$ upstream of the turbine. Note, that the stability parameter was measured to $\zeta = -0.089$ at $z = 10$ m, which at hub height corresponds to $\zeta_{\mathrm{ref}} = -0.29$. Also, the streamwise turbulence intensity was measured at hub height to $I_{u,\mathrm{ref}} = 12.6$ %, which is converted to the total turbulence intensity as $I_{\mathrm{ref}} \approx 0.8 I_{u,\mathrm{ref}}$ (van der Laan et al., 2015b). This conversion could actually be slightly different in the unstable ASL, because the ratios of velocity variance change with stability (e.g. Chougule et al., 2018), but unfortunately only the vertical velocity variance follows MOST, so that no general surface layer formula can be constructed (Panofsky and Dutton, 1984; Wyngaard, 2010). The operational state parameters $\{C_T, P, \Omega\}$ were taken from the OpenFAST steady-state curves, which were supplied for the benchmark.

For the unstable SWiFT case, the wake profile was only measured at $3D$ downstream and the results are shown in Fig. 10. Three different LES codes were used in the benchmark and the filled area in Fig. 10 represents the spread of the LES results. It can be seen that all three LES's underpredict velocity deficit compared to the LIDAR measurements, which highlights the





fundamental problem of comparing measurements with numerical models: Even highly computational expensive simulations
do not always match experimental results. This must either be due to experimental errors in the provided input data or because
the idealizations used for the LES's are too simple to capture the wake behaviour.

RANS can generally not be expected to perform better than a well-performed LES and if it does, it is likely due to fortunate
error cancellations. Therefore, from a theoretical point of view one could argue that the performance of RANS should mainly

be assessed with regards to how well it matches the LES results. Both RANS and LES use many of the same idealizations
(uniform roughness, flat terrain, homogeneous inflow, etc.) and indeed our RANS results in Fig. 10 are also closer to the LES
results, than to the experimental results.

For the SWiFT case the 2017 model seemingly performs better than the cstB model, but this is probably due to fortuitous
model error and/or some unaccounted mesoscale effects. Model error was expected, because the neutral EllipSys3D RANS

simulation in the SWiFT study (surprisingly) did not compare well with experimental results (Doubrawa et al., 2020), despite
EllipSys3D having been validated by van der Laan (2014) for numerous neutral cases. The very low wind veer of the stable
SWiFT case (c.f. Doubrawa et al., 2020) indicates that mesoscale effects were present during the experimental campaign.

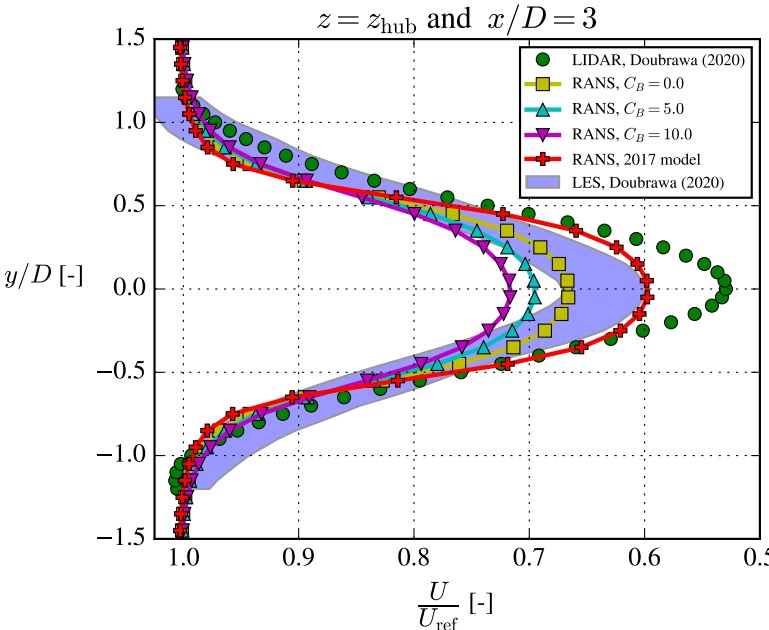

**Figure 10.** The unstable SWiFT case. The "fixed frame of reference" experimental and LES results were digitized from Doubrawa et al. (2020).

### 4.2 NTK41 case

A Nordtank NTK41 500kW wind turbine was installed at what is now the Risø campus of the Technical University of Denmark

in 1992, and was used for many research studies before its decommissioning in 2021. Among these studies, the "NTK41





testcase" of this paper is based on LIDAR measurments and LES's conducted by Machefaux et al. (2016). They used two different models for their LES's; the results included in Fig. 11 (along with our RANS results) are from their more advanced model, which they called the "LES-ABL" or "extended model." The inflow parameters ("NTK41" row in Table 2) and LIDAR measurements were ensemble-averaged over 20 10-min averages.

The NTK41 turbine is a stall-regulated wind turbine and is therefore operated at constant rotational speed independent of the inflow wind speed (Hansen, 2015), in this case at $\Omega = 27.1$ rpm. The thrust coefficient for the unstable case of Machefaux et al. (2016) was measured with strain-gauges to be $C_{T,meas} = 0.71$, while their LES gave $C_{T,LES} = 0.83$. Looking up the thrust curve of the NTK41 turbine at $U_{ref} = 6.8$ m/s also gives $C_{T,curve} = 0.83$, so this will be used in the present RANS simulations. They argued that the lower thrust coefficient of their measurement could be explained by the large uncertainty of 290    the strain gauges. Finally, the measured power was $P_{meas} = 120$ kW, while $P_{LES} = 127$ kW and $P_{curve} = 125$ kW, where the latter will be used to set $C_P$ for the AD model of the RANS simulations.

Figure 11 shows that the cstB model matches the LES and experimental data better than the 2017 model, although a still faster wake recovery is seen for both of these reference data. Compared to more conventional LES setups (e.g. V80-Abkar, V80-Keck and NREL5MW cases), the LES model used by Machefaux et al. (2016) is simplified by using a modified Mann box 295    for its inflow, and a slip condition at the bottom wall; together these add some uncertainty to the LES results. The experimental wake data can also be expected to have large uncertainties and/or biases connected to them, but the sources and sizes of these were not discussed in detail by Machefaux et al. (2016).

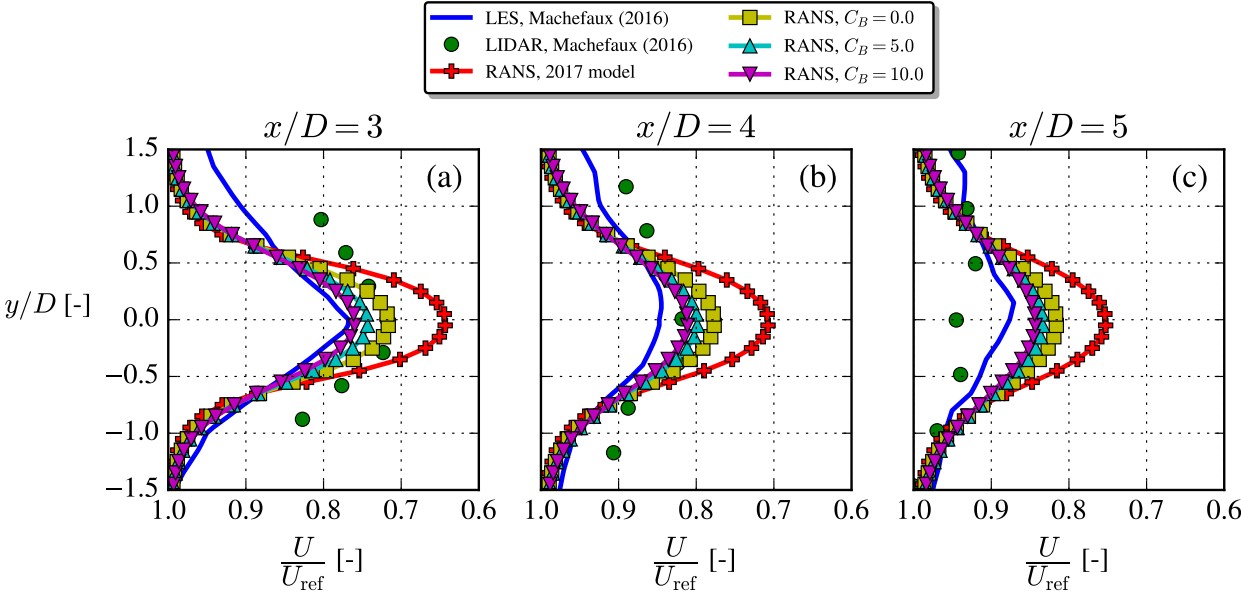

**Figure 11.** The unstable NTK41 case, where LES and experimental results were digitized from Machefaux et al. (2016). Profiles extracted at $z = z_{ref}$.





### 4.3 V80-Abkar case

Abkar and Porté-Agel (2014) investigated the effect of atmospheric stability using LES and used the results to modify the
analytical Bastankhah wake model (Abkar and Porté-Agel, 2015). They studied the wake of a single Vestas V80 turbine (known
from e.g. the Horns Rev 1, North Hoyle and Princess Amalia wind farms), which has been used in many previous wake studies.
The turbine was modelled in their studies by an airfoil-AD and operated at $\Omega = 16.1$ rpm. Neither $C_T$ nor $P$ were mentioned
in the two papers, so in the following RANS simulations the values deduced from the power and thrust coefficient curves of
Hansen et al. (2012) evaluated at $U_{\mathrm{ref}} = 8$ m/s, were used: $C_T = 0.81$ and $P = 696$ kW.

Relative to the LES, the velocity deficits shown in Fig. 12 are overpredicted by the 2017 model for all three downstream
distances, while the cstB model corrects this especially well in the far-wake. Besides the velocity deficit, the TI based on the
freestream velocity was also available for this case and is plotted in the lower row of Fig. 12 with the RANS results. The wake
TI is overpredicted by RANS, which is also typically seen for neutral RANS simulations (van der Laan, 2014).



**Figure 12.** The unstable V80-Abkar case, where LES results were digitized from Abkar and Porté-Agel (2015). Profiles extracted at $z = z_{\mathrm{ref}}$. Note, total TI (lower row) is based on $U_{\mathrm{ref}}$ and not the local velocity.

## 4.4 V80-Keck case

This case is based on a LES from Keck et al. (2014), where the SOWFA solver was used with a similar setup as in the work of Churchfield et al. (2012), i.e. using a precursor simulation for the inflow and modelling the turbine with Actuator Lines (AL). More specifically the "unstable North Hoyle row A" case is considered here; it features four V80 turbines spaced $11D$ apart, using the inflow parameters described in the "V80-Keck" row of Table 2. Wake data is available at $x/D = \{4, 5, 6\}$ downstream of the first turbine and the induction effect of the downstream turbines on this data should therefore be minimal, hence they

are omitted from the RANS simulations. The inflow wind speed is $U_{\mathrm{ref}} = 8 \ \mathrm{m/s}$, the same as in the V80-Abkar case, thus the





same operational state of the wind turbine is utilized, c.f. Table 2. The streamwise TI given by Keck et al. (2014) is converted to total TI with $I_{\text{ref}} \approx 0.8 I_{u,\text{ref}}$, similar to the method used in the SWiFT case.

Figure 13 shows that the cstB model improves the velocity deficit prediction over the old 2017 model, when comparing with the LES results, which were digitized from Fig. 11 in Keck et al. (2014) (note that a typo is present in that figure, i.e. label should be "[D]" instead of "[R]"). Streamwise TI was also available at the same downstream distances and in the RANS simulations it was obtained by converting from the total TI as described above. In both the streamwise TI and velocity deficit LES data, a misalignment of the wake center can be observed, which Keck et al. (2014) explains with that only 10 min of LES data was averaged. This is especially visible in the streamwise TI plots, but nevertheless it seems that the cstB model predicts streamwise TI in the right range except for an overprediction in the wake center, which was also seen in the previous case.

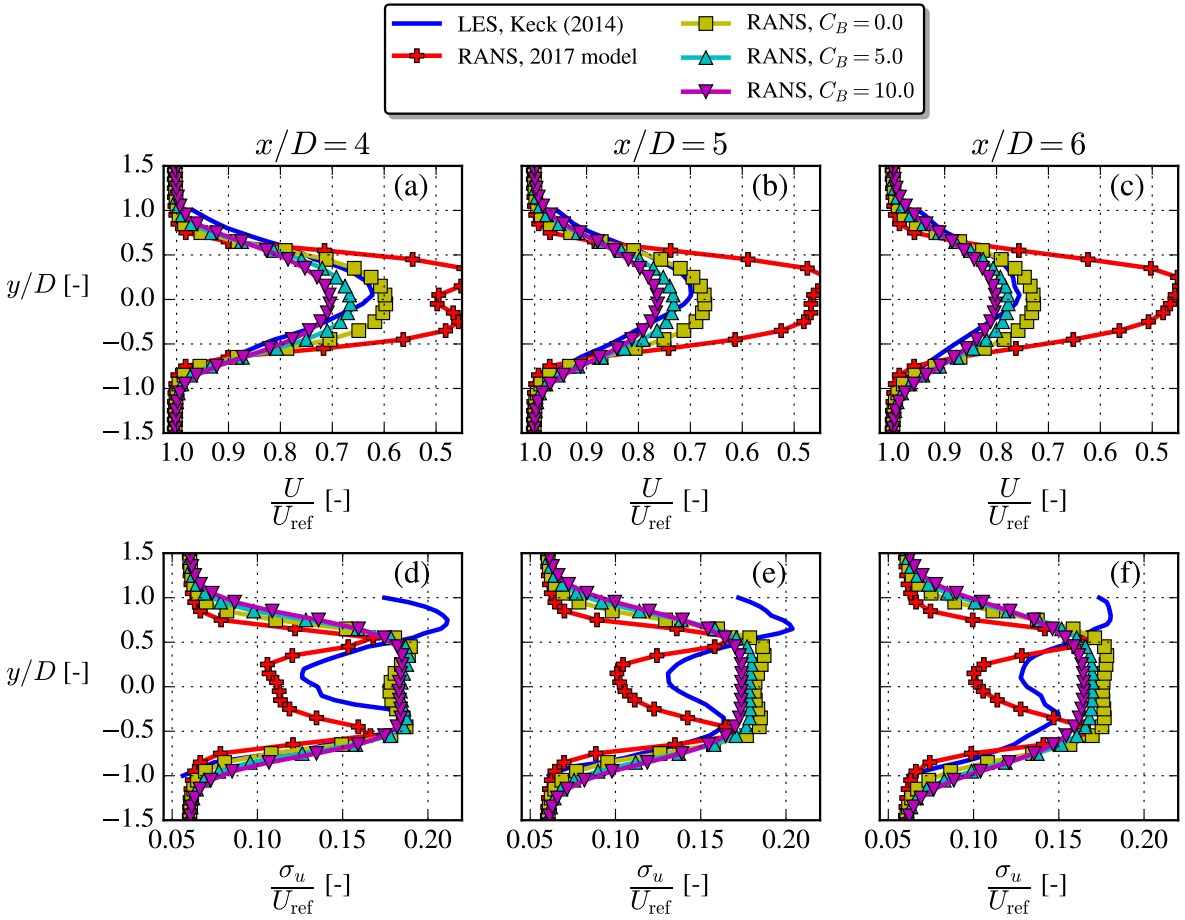

**Figure 13.** The unstable V80-Keck case, where LES results were digitized from Keck et al. (2014). Profiles extracted at $z = z_{\text{ref}}$. Note, streamwise TI (lower row) is based on $U_{\text{ref}}$ and $\sigma_u \approx \sqrt{\frac{2}{3}k}/0.8$ for the RANS simulations.





### 4.5 NREL5MW case

The last case is based on the LES studies by Churchfield et al. (2012), more specifically their "U-L case" (see inflow parameters in the "NREL5MW row" of Table 2). They model two NREL5MW turbines with the Actuator Line (AL) methodology coupled to the aeroelastic FAST solver and the turbines are spaced $7D$ apart. For the RANS simulations of the present study, we omit the second turbine and only compare with the first wake of the LES study. To avoid biases from the induction zone of the second turbine, we only consider wake results $\geq 2D$ upstream of the second turbine.

The steady-state power, thrust coefficient and rotational speed were not given in the paper, so therefore the steady-state curves from the DTU in-house aeroelastic solver, HAWCStab2, were used. These are similar to the curves shown by Jonkman et al. (2009), except that the thrust of Jonkman et. al. also includes gravity and therefore can not be used to obtain the aerodynamic thrust.

The velocity deficit of the new cstB model compares well with the LES data, especially so for $C_B = 5.0$.

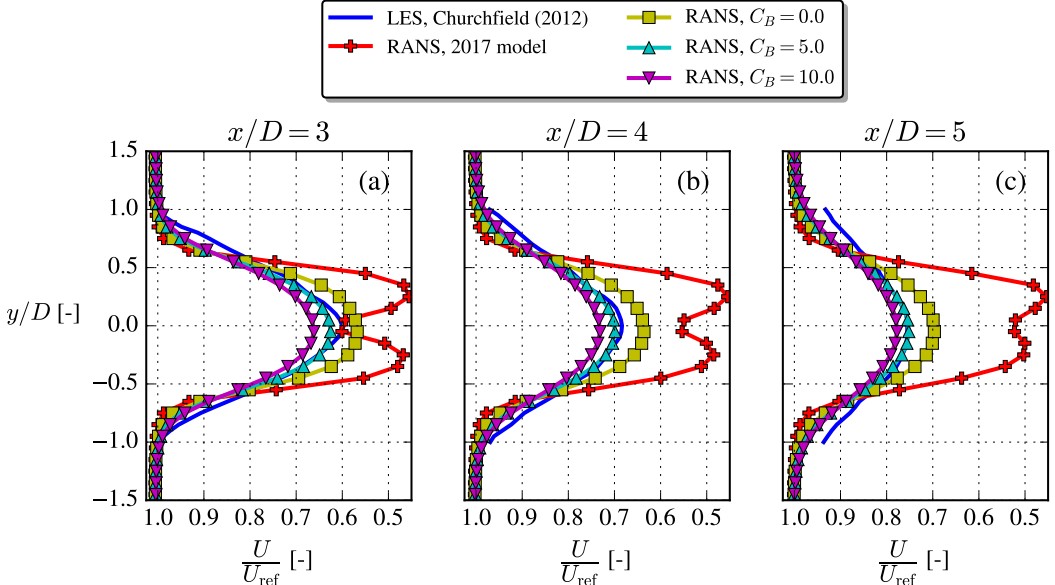

**Figure 14.** The unstable NREL5MW case, where LES results were digitized from Churchfield et al. (2012). Profiles extracted at $z = z_{\mathrm{ref}}$.

## 5 Conclusions

We have proposed a simple $k$-$\varepsilon$ RANS model, the "cstB" model, for simulation of wind turbine wakes in the unstable surface layer. The model does not require an additional temperature equation and instead bases the TKE buoyancy production on MOST and the assumption that it is decoupled from the wake dynamics, which means that the buoyant production of TKE is constant throughout the domain, even in the wake region, hence the name "cstB". Wind tunnel studies and simulations have



hinted that the latter assumption is reasonable, but a more thorough investigation would be beneficial for developing simple, non-neutral wake models.

Originally developed to account for the general over-diffusive nature of $k$-$\varepsilon$ models in wind turbine wakes under neutral conditions, here the $f_P$-correction is combined with the new cstB model by making two non-neutral modifications. These

345    introduce a new parameter, $C_B$; it is a free parameter analagous to $C_R$ in the original $f_P$-formulation. Both modifications are consistent in the sense that the new, non-neutral $f_P$-formulation becomes equal to the original neutral $f_P$ form for $\zeta \to 0$. By using this updated $f_P$-model with the cstB model, a faster wake recovery is obtained for unstable conditions over neutral conditions, when TI is fixed, as was also the case when no $f_P$-model was applied.

The cstB model with the modified $f_P$-function was generally found to perform better than the previous model of van der

350    Laan et al. (2017) with the old $f_P$-formulation, in terms of velocity deficit profiles from five different reference cases found in studies from the literature. Based on these comparisons, we recommend $C_B = 5.0$ to be used, but also acknowledge that each reference case were originally conducted with different numerical and experimental setups, and that further studies are needed to conclude on a more certain $C_B$ value, which could also be slightly code-dependent, as has been seen for $C_R$ in the original $f_P$-model.



# Appendix A: Simulation details

## A1 Grid study

Earlier studies by van der Laan et al. (2015b) have shown that a domain resolution of eight cells pr. diameter is sufficient to obtain grid independence for wakes in the neutral ASL. A range of domain and AD resolutions are here tested for the new cstB model and $f_P$-modifications with the TI = 12 % and $C_B = 5.0$ case also used in Fig. 9. The domain size is described in Fig. 3 and the Joukowsky-AD (see next section) is used. Disk averages of velocity and TI are evaluated $1D$ downstream of the turbine to verify grid independence in Fig A1.

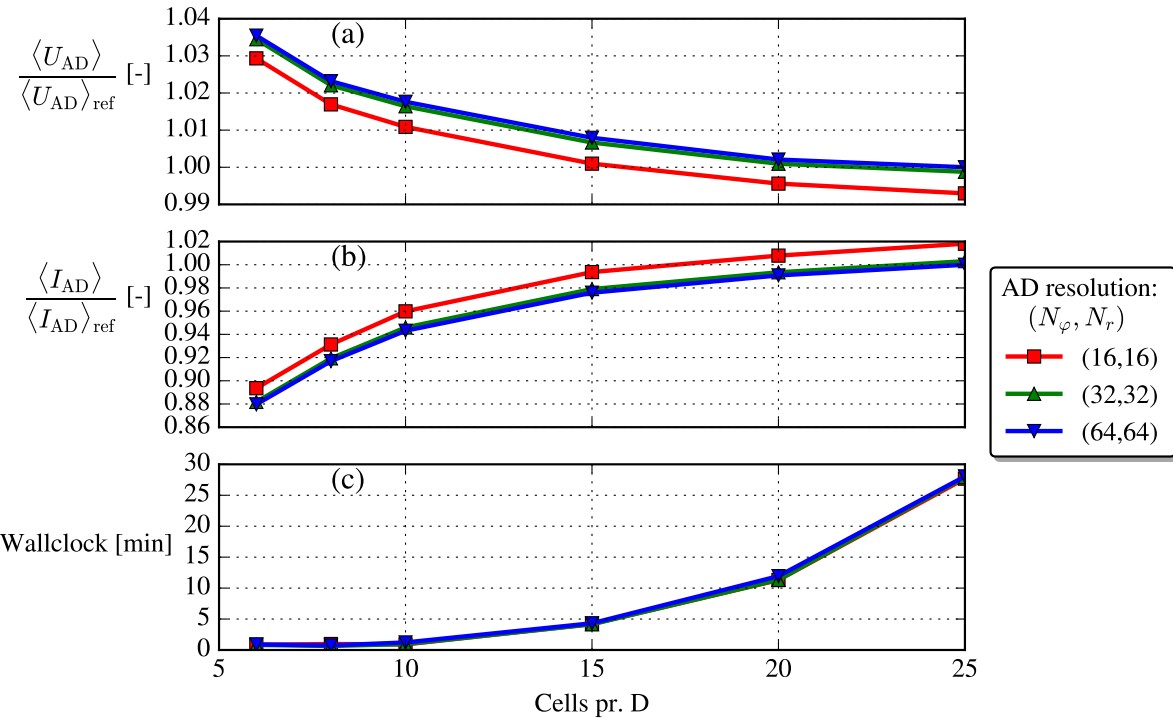

**Figure A1.** Grid-independence study. "Ref" = reference, i.e. the finest resolution available. Metrics are evaluated at $x/D = 1$. The cores used for the increasing domain resolutions are $\{45, 54, 63, 54, 60, 57\}$, respectively (non-constant, because the domains are decomposed in different number of blocks).

Based on this small grid study, a domain resolution of 10 cells pr. diameter and an AD resolution of $(N_\varphi, N_r) = (32, 32)$ is chosen for the current investigation. The difference between this resolution and the reference resolution is less than 2 % for the velocity metric and less than 6 % for the TI metric (the differences decrease with downstream position, e.g. at $x/D = 5$ the differences are only approximately 0.2 % for the velocity metric and 2 % for the TI metric). A simulation with this choice can be executed in about one wallclock minute on 63 cores (AMD EPYC 7351 processors are used).



## A2 The Joukowsky AD method

In summary, the surface force distributions (unit: $\mathrm{N/m^2}$) on the AD are calculated in each iteration as:

$$f_{n,ij} = \frac{\frac{1}{2}\rho C_T A U_{\mathrm{ref}}^2}{F_n^0} f_{n,ij}^0 \qquad f_{n,ij}^0 = 4\rho q_0 \frac{g(\chi_i)F(\chi_i)}{\chi_i}\left(\lambda\chi_i + \frac{1}{2}q_0\frac{g(\chi_i)F(\chi_i)}{\chi_i}\right)\frac{U_{ij}^2}{\left(1+\sqrt{1-C_T}\right)^2} \tag{A1}$$

$$f_{\theta,ij} = \frac{\frac{1}{2}\rho C_P A U_{\mathrm{ref}}^3}{P^0} f_{\theta,ij}^0 \qquad f_{\theta,ij}^0 = 2\rho q_0 \frac{g(\chi_i)F(\chi_i)}{\chi_i}\frac{U_{ij}^2}{1+\sqrt{1-C_T}} \tag{A2}$$

$$q_0 = \frac{\sqrt{16\lambda^2 a_2^2 + 8a_1 C_T} - 4\lambda a_2}{4a_1} \qquad a_1 = \int_0^1 \frac{g^2 F^2}{\chi}d\chi \qquad a_2 = \int_0^1 gF\chi d\chi \tag{A3}$$

$$g = 1 - \exp\left[-a\left(\frac{\chi}{\overline{\delta}}\right)^b\right] \tag{A4}$$

$$F = \frac{2}{\pi}\arccos\left[\exp\left(-\frac{N_b}{2}\sqrt{1+\lambda^2}\cdot(1-\chi)\right)\right]. \tag{A5}$$

Here, $f_{n,ij}$ and $f_{\theta,ij}$ are the normal and azimuthal surface force distributions at the $(i,j)$'th AD element ($i$: radial direction, $j$: azimuthal direction), which are applied in the CFD domain using the methodology described by Réthoré et al. (2014) and Troldborg et al. (2015). $F_n^0 \equiv \sum_i\left(\sum_j\left[f_{n,ij}^0 A_{ij}\right]\right)$ is the total normal force of the unscaled distribution, $P^0 \equiv U_{\mathrm{ref}}\lambda\sum_i\left(\chi_i\sum_j\left[f_{\theta,ij}^0 A_{ij}\right]\right)$ is the total power of the unscaled distribution, $\lambda \equiv \Omega R/U_{\mathrm{ref}}$ is the tip-speed ratio, $A$ is the area of the AD, $A_{ij}$ is the area of the $(i,j)$'th AD element, $\chi_i \equiv r_i/R$ is the local normalized radius, $U_{ij}$ is the normal velocity at the $(i,j)$'th AD element, $N_b = 3$ is the number of blades, $q_0$ is the normalized circulation, $g$ is the root correction and $F$ is the tip correction. The latter two are obtained with Delery's root correction (with parameters $a = 2.335$, $b = 4.0$ and $\overline{\delta} = 0.25$) and Prandtl's tip correction, respectively, c.f. Eq. (A4) and (A5).

The normal and tangential loadings for the same case as used in Section A1 are compared between the uniform-AD, airfoil-AD and Joukowsky-AD, the latter with two different $\overline{\delta}$'s, in Fig. A2. Clearly, the Joukowsky-AD with $\overline{\delta} = 0.25$ produces similar loadings as the airfoil-AD, e.g. qualitatively correct root behaviour, tip behaviour and constant tangential loading region, which makes it superior to the uniform-AD.

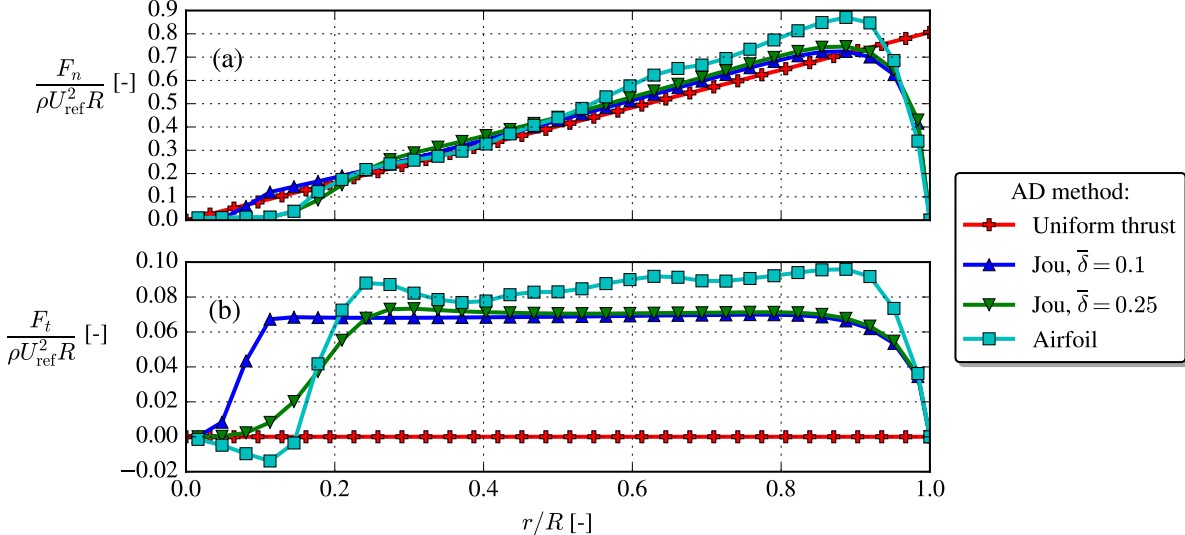

**Figure A2.** The normal blade loading, $F_n$ [N/m], and tangential blade loading, $F_t$ [N/m], are normalized by the density, $\rho$, rotor radius, $R$, and freestream hub height velocity, $U_{\text{ref}}$. The blade loadings for the Joukowsky-AD and airfoil-AD have been obtained by azimuthal averages , while $F_n = \frac{2\pi r T/A}{N_b}$ is prescribed a-priori for the uniform-AD.

Finally, in Fig. A3 the velocity and TI disk averages follow the same trend for all four AD methods, but with a slightly larger velocity deficit for the airfoil-AD, possibly because of its also slightly larger blade loadings, see Fig. A2. The thrust coefficient of the uniform-AD and Joukowsky-AD is $C_T = 0.77$, which from 1D momentum theory should give $U/U_{\text{ref}} = $

$1 - 0.5\left(1 - \sqrt{1 - C_T}\right) \approx 0.74$ at the rotor plane. This is not exactly observed in Fig. A3, but contrary to ideal 1D momentum theory our CFD simulation also includes atmospheric turbulence and shear.





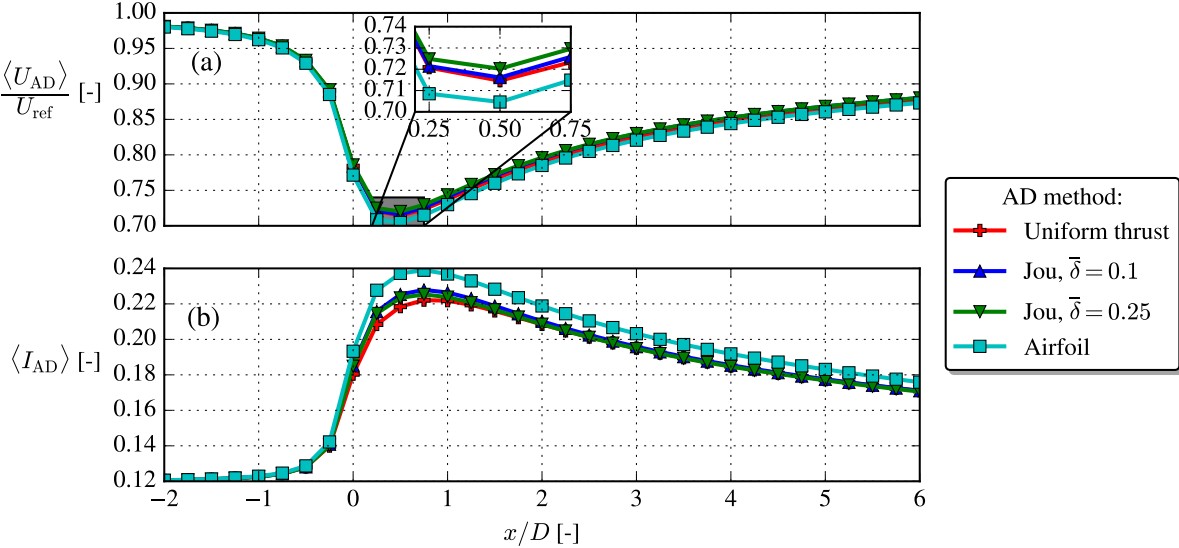

**Figure A3.** Disk average of velocity, $\langle U_{\mathrm{AD}} \rangle$, and disk average of turbulence intensity, $\langle I_{\mathrm{AD}} \rangle$.

*Code availability.* `EllipSys3D` and `pywake_ellipsys` are propriatary software of DTU. Information about the latter can however be freely accessed at https://topfarm.pages.windenergy.dtu.dk/cuttingedge/pywake/pywake_ellipsys/index.html.

*Data availability.* The RANS results were generated with DTU's proprietary software, but the data presented can be made available by
contacting the corresponding author. Interested parties are also welcome to hand-digitize the results and use them as reference in other publications.

*Author contributions.* MB performed the RANS simulations and proposed the modifications to the turbulence model. All authors (MB, MPvdL and MK) contributed to derivation of the new model and article writing.

*Competing interests.* The authors declare that they have no conflict of interest.

*Acknowledgements.* This work has been carried out under the Poul la Cour fellowship. The authors are grateful to the creators of the LES and LIDAR results used in the validation section.





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
