# Peer review of "RANS modelling of a single wind turbine wake in the unstable surface layer"

_Wind Energy Science, 2021_

## Referee Comment (RC2)

Review : RANS modelling of a single wind turbine wake in the unstable surface layer

The present paper deals with the modelling of wind turbine wakes in the unstable atmospheric boundary layer using a RANS approach. Two models are proposed: the first on aims at accounting the buoyant production of TKE without relying on a temperature equation. The second model improves the so-called k-epsilon-fp RANS turbulence model, based on observed discrepancies and inconsistencies against experimental data and higher-fidelity simulations, i.e. LES. The model is globally clear and very well written.

I would like to start this review with a general discussion about the proposed approach. It is my understanding that in an unstable ABL, the faster wake recovery (with respect to neutral conditions) that is observed can be attributed to the large levels of wake meandering that smear out the wake. The meandering itself is due to a large amount of lateral (y-wise) turbulence intensity in the ABL. In other words, it seems difficult to me to neglect the anisotropic nature of the ABL when dealing with wind turbine wakes. I think it is necessary to include a discussion on that topic in the paper and explain how the authors think they can deal with such anisotropic flows using a two-equation turbulence model, based on an isotropic turbulence assumtion. I understand the main objective is to propose an efficient, intermediate fidelity model (i.e. in between analytical and LES), but the necessary physics should be there and properly represented.

The authors introduce a new so-called "cstB" model to account for buoyant TKE production. Although the reasoning is clearly explained, there is, from my point of view, a major drawback in this paper: the model, that seems to make sense physically speaking, is never quantitatively validated, and thus none of the assumptions are properly justified. Only a brief qualitative discussion is provided. It is surprising, since two cited references (Zhang et al. 2013, Hancock and Zhang 2015) contain quantitative data that could be used for validation purpose, if I am correct. Furthermore, a minimal validation/verification of the model consistency, that should be provided, is a comparison to the flux-gradient approach (section 3.1), with the integration of the temperature equation in the system. I guess this is feasible with Ellipsys and should be integrated for comparison/validation purposes.

It is my point of view that these drawbacks also apply to the presented modifications to the k-epsilon-fp model. Some improvements are introduced. These are mainly based on mathematical consistency (i.e. neutral ABL limit) but are not properly justified (no real physical explanations are provided). And, in the end, the proposed validation cases focus on "global" flow properties such as wake velocity deficit or TKE levels. It is my opinion that this paper would gain a lot by showing proper comparisons to LES simulations (or experimental data): one might be able to extract the dissipation rate, the eddy-viscosity (and then estimate fp through (16)), or other quantities that would help to asses the pertinence of the choices that are made. See P.E. Réthoré PhD Thesis as a typical example.

About the presented results:

- the TKE levels appear to be overall over-estimated, while the velocity profiles match well. Can the authors provide some analysis on this inconsistency?

- Only single-wake results are compared with LES and/or experiment, as indicated in the paper title. However, it seems that some of the presented LES are based on multi-turbine simulations… Which may rise some doubt about the capacity of the model to deal with wake superposition. Does it perform well in such cases, as for the single wake cases?

About the grid convergence study: wake TI and velocity profiles are extracted 1D behind the wind turbine. However, the presented validation results are taken at 3, 4, 5… up to 12D. And I have the feeling it is easier to converge at 1D than at 12D, since there are much less diffusion effect. What is the reason for this choice? I personally think it is necessary to include the profiles at, let's say, 6 and 12D behind the wind turbine.

One last general remark: can this model be adapted to stable conditions? And if already done, how does it perform? I guess it is the authors objective to end-up with a model that is valid for all the classical ABL stability a wind turbine may encounter.

In the end, both models seem to lead to improve previous modelling approaches, and it sounds like a step forward is achieved in this paper. Thus, I strongly encourage the authors to provide more justification and adapted validation to their work. This will surely provide some confidence in the proposed approaches, although I am aware having high quality data for such unstable cases is not that easy. Most probably running and comparing the RANS approach with an LES simulation would help.

---

## Author Comment (AC1)

**Reply to reviewers**

December 22, 2021

We would like to thank the two reviewers for undertaking the job of reviewing our work. All the detailed feedback and suggestions are much appreciated, and have contributed to the value as well as the clarity of the paper. The reviewers comments are copied and answered (blue color) in this document, while an additional document is provided that highlights all modifications with respect to the initial submitted version.

**Reviewer 1**

The article "RANS modelling of a single wind turbine wake in the unstable surface layer" by M. Baungaard et al. proposes new derivation of a k-epsilon RANS model for unstable atmospheric boundary layer applied to a single wind turbine. After a very clear introduction, the general simulation set-up is given in Section 2 with velocity, k and epsilon inflow profiles, wind turine modeling and RANS parameters. In the latter, the van der Laan el al. (2017) modified model is described. In Section 3, the derived k-epsilon model is presented with the fP-limiter parameter. It mainly consists in two modifications: a constant formulation of the buoyant production of Turbulent Kinetic Energy (correcting the van del Laan (2017) unphysical effects) and a new formulation of the fP parameter (via modifications on f0 and CR parameters). This model is applied on 5 cases in Section 4 and compared to experimental or LES results, showing a better agreement of the new model compared to the one of van der Laan et al. (2017). The article is very clear on its objectives, the new methodology proposed and the validation process and is very well written. Some informations may lack of precision or justification (given below) but the overall article is well justified and deserve to be published and adressed to the wind energy community.

**Specific comments:**

1. Sec 1, Introduction, line 19: although precised later in the text, the RANS model can also include complex gemometry that analytical models cannot .

   Indeed. Adopted.

2. Sec 2, Simulation set-up, line 62: the numbers given are from an article of 2015. As the computational ressources evolve very quickly, are these numbers still relevant? .

   The time for a wind turbine wake RANS simulation ($\sim 1$ wallclock minute at $\sim 50$ CPU's) is contemporary, see Appendix A in our article. While we don't simulate any LES ourselves in this paper, our experience for recent similar LES runs in EllipSys3D (not published) is that LES is still on the order of $10^3$ more expensive than RANS in terms of CPU-hours as was also reported by van der Laan et al. (2015). The question of computational costs is naturally heavily dependent on the CFD code, turbulence model, mesh, CPU, user experience, etc., so we would like to stress that $10^3$ is a rough estimate.

3. Sec 2, line 66: the comparison in terms of computational ressources needed and return time between LES and RANS is well described and objective. Is a comparison with the engineering models would also be relevant here?

We do not think of an engineering model as a "simulation tool", which is why we only discussed the cost of RANS and LES in Section 2 (section 2.1 is called "simulation setup"). If an engineering wake model is programmed in an efficient manner, it typically takes on the order of seconds or milliseconds to execute on a regular laptop, so there is essentially no computational cost.

4. Sec 2.1, line 94: the reason to not take into account more realistic inflow profiles covering the whole ABL is understandable. But what would be the implications or consequences in terms of physics (compatibility of the proposed model for example) or numerics (impact computation time for example)?

   This is a topic we have also been considering, and wish to pursue in the future; hopefully we can give a good answer with regards to the implication of more complex inflow profiles in a future article. A range of more realistic inflow profiles from the "ABLp" and "ABLc" models are discussed by van der Laan et al. (2021a), but they are only available for neutral/stable conditions.

5. Sec 2.1, line 96: the choice of K and Cu values may be justified

   The $\kappa$ and $C_\mu$ values have been moved to Table 2.1 and a reference to Sørensen (1995) is added.

6. Sec 2.1, line 116 + Fig. 2: the comments on the eddy viscosity decomposition are poor and may be expanded

   We agree and have re-written the section. The message we were trying to send with the decomposition and figure is that the faster wake recovery in typical unstable conditions is not only caused by increased velocity scale (i.e. TI), but also due to an increased turbulent length scale.

7. Sec 2.2, line 124: the advantage of the Joukowsky-AD compared to airfoil-AD is not very clear. Is it just interesting because it uses few input parameters?

   Yes, exactly. The airfoil-AD requires information about chord-distribution, twist-distribution, airfoil-distribution and airfoil data (lift- and drag-coefficients for a wide range of angle of attacks) for each airfoil type. The Joukowsky-AD requires none of these, but still predicts similar results for the force distributions on the disk, Appendix B.

8. Sec 2.3, line 143 + Fig. 3.: the vertical mesh stretching chosen coupled to the large domain should imply a large number of mesh cells out of the wake zone, i.e. the interesting area. This is a classical drawback of stretched structured grids. Can you give the number of cells into the wake region (the one of Fig. 3.) over the total number of elements and comment?

   The number of cells in the z-direction between $0 < z/D < 3$ is $n_z = 70$ (not written in the article), while one can calculate $n_x = \frac{l_x}{\Delta x} = 160$ and $n_y = \frac{l_y}{\Delta x} = 40$. The total amount of cells was given above Fig. 3 in the paper $N_{total} = 2.1$ million, hence the requested ratio is: $\frac{n_x n_y n_z}{N_{total}} \approx 0.22$. It is therefore correct that the majority of the cells are outside of the region of interest, which certainly is drawback in terms of computational cost (this has been added to the paper). The drawback of *not* using a large domain is artificial tunnel-blockage and streamwise developing inflow profiles (even with the $S_k$ modification of the $k$-equation there will be some initial development). Some studies, e.g. van der Laan et al. (2021a), use even larger domains of size 100 km, while others only simulate in the wake domain, and in this paper we choose a compromise of 10 km.

9. Sec 2.3, line 163: the Coriolis force and veer effect are not taken into account. Can you briefly justify?

   It is of course more physically correct to retain the Coriolis term, but removing it makes the model Reynolds-number similar; this is a nice feature for wind farm simulations, because it can be used to simulate a range of cases with different inflow speeds in a more efficient manner (cost goes down by a factor of 2 to 3 for a typical wind farm AEP calculation according to van der Laan et al. (2021b)). Unstable conditions are typically charecterized by little veer, so this can also to some extend justify the choice.

10. Sec 3., Fig 4: What is nu_tref ? Why the viscosity ratio drops towards zero in the rotor area and in the near wake? This figure may be more discussed.

    $\nu_{\text{tref}}$ is the freestream kinematic eddy viscosity at hub height (clarified in the figure caption now). A standard $k$-$\varepsilon$ model is known to be unrealizable in the rotor area and near wake regions, which leads

to an overestimation of the wake recovery (Réthoré, 2009); the $k$-$\varepsilon$-$f_P$ corrects this behaviour by using $f_P < 1$ in regions with large velocity gradients leading to $0 < \nu_t/\nu_{\text{tref}} < 0.4$ (it does not go exactly to 0, but to some value in this range according to the colorbar) in the two aforementioned regions.

11. Sec 3.1, Fig. 7: the comments on Fig. 7 are poor. More discussions can be added on wake recovry via velocity field, on shear parameters or turbulent time.

    Agree, we have added discussion/motivation of the shear parameter. The turbulence time scale is removed, since it is not of use for the current discussion.

12. Sec 3.2, Fig. 8 & 9: Same remark

    Fig. 8 has been simplified and the small discussion for Fig. 8 and 9 has been re-written.

13. Sec 4., line 243: the choice of CB and CR values is unclear. As CR depends on CB, how can CR=4.5 be fixed ?

    The value of $C_R$ is *not* fixed to 4.5, we simply write that this is the value of $C_R$ in the neutral limit, i.e. when $\mathcal{B}/\varepsilon = 0$. As written around the text of eq. 28, $C_B$ is a new parameter to be calibrated and we choose/recommend $C_B = 5$ from an assessment of the validation cases in Sec. 4. We agree that a more rigorous statistical approach (e.g. similar to the calibration procedure by van der Laan et al. (2015)) would be better, but the quality and variety of our reference data is in our opinion not feasible for such an analysis. It is therefore only a "loose" recommendation of $C_B = 5$, which we also try to underline in the conclusion of the paper.

14. Sec 4., line 248: are 5 applications needed? Some case give the same insights (V80-Abkar and V80-Keck for example).

    One can both argue there should be less cases (to simplify the paper) and that there should be more cases (the more cases, the more fair validation). The V80-Abkar and V80-Keck cases both simulate the V80 turbine, but using different numerical codes and inflow conditions, so in our opinion their are both valuable sources for validation.

15. Sec 4.3 & 4.4, Fig. 12 & 13: The overprediction of TI of the proposed model is observed, while the 2017 model behaves better in near wake. Can you explain why?

    There are only two cases with TI (V80-Abkar and V80-Keck), so we should be careful about making general conclusions from these, although the 2017 model indeed seems to predict near-wake TI better than the cstB model. As also mentioned in the V80-Abkar case description, RANS models typically overestimate TI, which is also seen for the cstB model. The 2017 model has a slower wake recovery in terms of velocity deficit, which would be consistent with also having less turbulence development in the near-wake region; the wrong behaviour of the 2017 model with regards to velocity deficit therefore seems to counteract the general TI problem with RANS models. We prioritize predicting a correct velocity deficit, since this metric is used for AEP calculations.

**Technical comments:**

1. line 36: Tubulent $\to$ Turbulent

   Adopted.

2. Table 1: Cu and K are already given in the text line 96. It shoudn't be repeated here.

   We have removed $C_\mu$ and $\kappa$ from the text; the constants are only given in Table 1 now.

3. Comma after Equations 9, 13, 14, 15, 16, 17, 18, 20, 22, 23, 24 and point afer Eq. (21). Move the point of Eq. 26 after the parenthesis

   Adopted.

4. Line 273: comma after "For the SWiFT case"

   Adopted.

**Reviewer 2**

The present paper deals with the modelling of wind turbine wakes in the unstable atmospheric boundary layer using a RANS approach. Two models are proposed: the first one aims at accounting the buoyant production of TKE without relying on a temperature equation. The second model improves the so-called k-epsilon-fp RANS turbulence model, based on observed discrepancies and inconsistencies against experimental data and higher-fidelity simulations, i.e. LES. The model is globally clear and very well written.

I would like to start this review with a general discussion about the proposed approach. It is my understanding that in an unstable ABL, the faster wake recovery (with respect to neutral conditions) that is observed can be attributed to the large levels of wake meandering that smear out the wake. The meandering itself is due to a large amount of lateral (y-wise) turbulence intensity in the ABL. In other words, it seems difficult to me to neglect the anisotropic nature of the ABL when dealing with wind turbine wakes. I think it is necessary to include a discussion on that topic in the paper and explain how the authors think they can deal with such anisotropic flows using a two-equation turbulence model, based on an isotropic turbulence assumtion. I understand the main objective is to propose an efficient, intermediate fidelity model (i.e. in between analytical and LES), but the necessary physics should be there and properly represented.

Indeed our model predicts $\overline{u'u'} = \overline{v'v'} = \overline{w'w'}$ in the freestream; this is the case for all turbulence models based on the Boussinesq hypothesis (Prandtl's mixing length, $k$-$\varepsilon$, $k$-$\varepsilon$-$f_P$, $k$-$\omega$, etc.), so the only way to include anisotropy is to abandon the Boussinesq hypothesis, e.g. use differential Reynolds stress models, algebraic Reynolds stress models, or machine-learning approaches. This is beyond the scope of this paper, but is certainly interesting to investigate further in the future (even the neutral ABL has anisotropic freestream turbulence according to Panofsky and Dutton (1984), Chapter 7).

Abkar and Porté-Agel (2015) showed with LES data that there is meandering in both stable, neutral and unstable conditions; also there is a non-negligible vertical meandering in all conditions (approximately 60-70% of the lateral meandering at $5D$, see their Figure 16). Our steady RANS simulation obviously don't capture the dynamics of the meandering, but it does predict larger turbulent scales for increasingly unstable conditions, which is the physical origin of the increased meandering according to Keck et al. (2014).

The authors introduce a new so-called "cstB" model to account for buoyant TKE production. Although the reasoning is clearly explained, there is, from my point of view, a major drawback in this paper: the model, that seems to make sense physically speaking, is never quantitatively validated, and thus none of the assumptions are properly justified. Only a brief qualitative discussion is provided. It is surprising, since two cited references (Zhang et al. 2013, Hancock and Zhang 2015) contain quantitative data that could be used for validation purpose, if I am correct. Furthermore, a minimal validation/verification of the model consistency, that should be provided, is a comparison to the flux-gradient approach (section 3.1), with the integration of the temperature equation in the system. I guess this is feasible with Ellipsys and should be integrated for comparison/validation purposes.

The wind tunnel data of Zhang et al. (2013) or Hancock and Zhang (2015) could have been chosen for the validation procedure, however we preferred real-scale data (either LES or experiments) to avoid scaling effects and because artificial atmospheric profiles generated in wind tunnels are less reliable.

It is indeed possible to use an active transport equation for temperature and the flux-gradient relationship $\overline{u_i'\theta'} = -\frac{\nu_t}{Pr_t}\frac{\partial \Theta}{\partial x_i}$ in EllipSys as the Reviewer suggests, but it will not be possible to obtain a steady-state solution, because heat will continuosly be added to the domain through the wall BC, while there is no sink of heat in the domain. One can then question, if our simple steady-solution of the unstable ABL is even relevant; we choose to interpret it as an ensemble average of many similar unstable conditions, which could be of value for estimation of AEP. For other usages, e.g. day-to-day forecasting, the cstB model can not be used.

More discussions and validation of the physical model assumptions will be undertaken in a planned TORQUE paper next year using new LES data (see last question in this reply).

It is my point of view that these drawbacks also apply to the presented modifications to the k-epsilonfp model. Some improvements are introduced. These are mainly based on mathematical consistency (i.e. neutral ABL limit) but are not properly justified (no real physical explanations are provided). And, in the end, the proposed validation cases focus on "global" flow properties such as wake velocity deficit or TKE levels. It is my opinion that this paper would gain a lot by showing proper comparisons to LES simulations (or experimental data): one might be able to extract the dissipation rate, the eddy-viscosity (and then estimate fp through (16)), or other quantities that would help to asses the pertinence of the choices that are made. See P.E. Réthoré PhD Thesis as a typical example.

Modification 1 comes directly from the paper of Apsley and Leschziner (1998), while we agree that the arguments for modification 2 are more hand-waving. It is simply a choice to make the $C_R$ parameter flow-dependent. In the future we wish to explore more advanced turbulence models, e.g. the explicit algebraic Reynolds stress models, to avoid such ad-hoc assumptions. As stated in the previous question, there is a LES/RANS comparison study on its way.

About the presented results

- the TKE levels appear to be overall over-estimated, while the velocity profiles match well. Can the authors provide some analysis on this inconsistency?

  See answer 15. to Reviewer 1.

- Only single-wake results are compared with LES and/or experiment, as indicated in the paper title. However, it seems that some of the presented LES are based on multi-turbine simulations... Which may rise some doubt about the capacity of the model to deal with wake superposition. Does it perform well in such cases, as for the single wake cases?

  The 2017 model with $f_P$ (but without the modifications of this paper) was used by van der Laan et al. (2021a) to simulate a row of ten aligned turbines and it performed poorly for unstable conditions, i.e. it had significantly slower wake recovery compared to a similar neutral case and no wake equilibrium seemed to be reached. Preliminary results for the same case run with the cstB model looks better, see Fig. 1, in the sense that it has faster wake recovery than the neutral reference case for the first 3-4 turbines, while the wake deficit is the same in the fully developed region of the wind farm.

  The cstB model is a new approach, so our intention was to focus solely on single wake cases in this first paper, but we have now included this aligned row case in Appendix A3. We also agree that the model should be tested for more complicated scenarios in the future, i.e. wind turbine rows, full wind farms, AEP calculations, complex terrain, etc.

[Figure]

Figure 1: Aligned row of 10 turbines with inflow similar to the Case 5 of van der Laan et al. (2021a).

About the grid convergence study: wake TI and velocity profiles are extracted 1D behind the wind turbine. However, the presented validation results are taken at 3, 4, 5... up to 12D. And I have the feeling it is easier to converge at 1D than at 12D, since there are much less diffusion effect. What is the reason for this choice? I personally think it is necessary to include the profiles at, let's say, 6 and 12D behind the wind turbine.

As was stated in a small parenthesis in Appendix B, we already did the same grid convergence analysis at $5D$ downstream; for our simulations it showed that grid convergence is much more critical in the near-wake than in the far-wake, possibly because of the large gradients in the former region, which is why we showed the plots at $1D$ downstream.

One last general remark: can this model be adapted to stable conditions? And if already done, how does it perform? I guess it is the authors objective to end-up with a model that is valid for all the classical ABL stability a wind turbine may encounter.

It is relatively straight-forward to use this model in stable conditions (one just has to use other expressions for $S_k$ and $C_{\varepsilon 3}$, see van der Laan et al. (2017)), but its physical validity and practical value are questionable: The height of a nightime/stable ABL is on the order of 100-200 m. The cstB model is only physical valid in the ASL, where $h_{ASL}/h_{ABL} \approx 0.1$, hence it would strictly only be valid in the first 10-20 m above the ground. Probably a more viable route for stable conditions, would be the so-called ABLc and ABLp models (van der Laan et al. (2020),van der Laan et al. (2021b)), which can model stable conditions and be used throughout the whole ABL including the capping inversion.

In the end, both models seem to lead to improve previous modelling approaches, and it sounds like a step forward is achieved in this paper. Thus, I strongly encourage the authors to provide more justification and adapted validation to their work. This will surely provide some confidence in the proposed approaches, although I am aware having high quality data for such unstable cases is not that easy. Most probably running and comparing the RANS approach with an LES simulation would help.

We agree that more detailed LES data could help to justify and validate the model; the LES module of EllipSys is currently not optimally suited for atmospheric flows with stratification, but we have teamed up with external partners, who are capable of running such simulations. A TORQUE paper on this matter is planned for next year.

**Own improvements**

- Improved colorbars of Fig.4 and 7.
- Simplified Fig.8.
- Appendix A3 with an aligned row case.
- Change "pywake_ellipsys" to "PyWakeEllipSys" in the code availability section.

[revised manuscript text omitted]